site
Manuscript prepared for J. Name
with version 2014/09/16 7.15 Copernicus papers of the LaTeX class copernicus.cls.
Date: 29 August 2018

# The effects of intercontinental emission sources on European air pollution levels

Jan Eiof Jonson[1], Michael Schulz[1], Louisa Emmons[2], Johannes Flemming[3],
Daven Henze[4], Kengo Sudo[5], Marianne Tronstad Lund[6], Meiyun Lin[7],
Anna Benedictow[1], Brigitte Koffi[8], Frank Dentener[8], Terry Keating[9], Rigel Kivi[10],
and Yanko Davila[4]

[1]Norwegian Meteorological Institute, Oslo, Norway
[2]National Center for Atmospheric Research Boulder, Colorado, USA
[3]ECMWF (European Centre for Medium Range Forecast)
[4]University of Colorado Boulder, Colorado, USA
[5]NAGOYA-U,JAMSTEC,NIES, Japan
[6]Center for International Climate and Environmental Research (CICERO) - Oslo, Norway
[7]Program in Atmospheric and Oceanic Sciences of Princeton Univeristy and NOAA Geophysical
Fluid Dynamics Laboratory, Princeton, New Jersey, USA
[8]European Commission, Joint Research Centre, Ispra, Italy
[9]U.S. Environmental Protection Agency
[10]Finnish Meteorological Institute, Sodankylä, Finland

*Correspondence to:* Jan Eiof Jonson (j.e.jonson@met.no)

**Abstract.** This study is based on model results from TF HTAP (Task Force on Hemispheric Transport
of Air Pollution) phase II where a set of source receptor model experiments have been defined,
reducing global (and regional) anthropogenic emissions by 20% in different source regions throughout
the globe, with main focus on year 2010. All the participating models use the same set of anthropogenic
emissions. Comparisons of model results to measurements are shown for selected European surface
sites and for ozone sondes, but the main focus here is on the contributions to European ozone levels
from different world regions, and how and why these contributions differ depending on model. We
investigate the origins by use of a novel stepwise approach combining simple tracer calculations and
calculations of CO and $O_3$. To highlight differences, we analyse the vertical transects of the mid
latitude effects from the 20% emission reductions.

The spread in model results increase from the simple CO tracer to CO and then ozone as the
complexity of the physical and chemical processes involved increase. As a result of non linear ozone
chemistry the contributions from non European relative to European sources are larger for ozone
compared to CO and the CO tracer. For annually averaged ozone the contributions from the rest of the
world is larger than the effects from European emissions alone, with the largest contributions from
North America and East Asia. There are also considerable contributions from other nearby regions to
the east and from international shipping. The calculated contributions to European annual average
ozone from other major source regions relative to all contributions from all major sources (RAIR –

Relative Annual Intercontinental Response) have increased from 43% in HTAP1 to 82% in HTAP2. This increase is mainly caused by a better definition of Europe, increasing emissions outside of Europe relative to those in Europe, and including nearby non European source now as external-to-Europe regions. European contributions to ozone metrics reflecting human health and ecosystem damage, mostly accumulated in the summer months, are larger than for annual ozone. Whereas ozone from

European sources peaks in the summer months, the largest contributions from non European sources are mostly calculated for the spring months when ozone production over the polluted continents starts to increase, while at the same time the lifetime of ozone in the free troposphere is relatively long. At the surface contributions from non European sources are of similar magnitude for all European sub regions considered, defined as TF HTAP receptor regions (north west, south west, east and south east

Europe).

## 1  Introduction

This paper is based on the HTAP model experiment phase 2 (HTAP2), where CTMs (chemical transport models) perform model sensitivity studies, perturbing the emissions in different world regions. TF HTAP (http://www.htap.org/) is organized under the auspices of the UNECE Convention

on Long-range Transboundary Air Pollution (LRTAP Convention) and reports to the Convention's EMEP Steering Body. The HTAP2 experiment is described in more detail in Galmarini et al. (2017) and in the HTAP2 work plan, posted on the HTAP2 web site www.htap.org. All models should use the same set of anthropogenic emissions, see Janssens-Maenhout et al. (2015).

In particular the experiments is set up to:

– Examine the transport of air pollution, including ozone and its precursors and particulate matter and its components (including black carbon), across the Northern Hemisphere.

– Assess potential emission mitigation options available inside and outside the UNECE region.

– Assess their impacts on regional and global air quality, public health, ecosystems, and near-term climate change.

– Promote collaboration both inside and outside the Convention.

HTAP2 is a follow up of the HTAP phase 1 model experiment (HTAP1). Results from HTAP1 has been described in a series of peer review papers, including (Casper-Anenberg et al., 2009; Fiore et al., 2009; Reidmiller et al., 2009; Jonson et al., 2010; Sanderson et al., 2008; Shindell et al., 2008), and in the the HTAP1 main report (TF HTAP, 2010). The HTAP1 model experiment showed that

intercontinental transport of ozone and ozone precursors could explain a large portion of the ozone over Europe, but results differed substantially between the models.

A large number of CTMs have uploaded their results to the HTAP2 database. This study is limited to those models that, in addition to the base run, as a minimum have uploaded their source receptor

calculations for ozone reducing all anthropogenic global emissions and European emissions by 20%. Seven of the models fulfil these criteria.

A large number of papers from HTAP2 have been published, in the ACP (Atmospheric Chemistry and Physics) Special issue: "Global and regional assessment of intercontinental transport of air pollution: results from HTAP, AQMEII and MICS"

The effects of intercontinental transport of ozone to North America is discussed in Huang et al. (2017), but no such study has so far been made for Europe based on the HTAP2 data set. In this paper we aim to enhance our understanding of the contributions to European ozone levels from European and non-European sources. In order to better understand the transport patterns between the continents we use a novel stepwise approach, starting with a simple CO like tracer using the CO anthropogenic emissions and a fixed decay rate of 50 days. As all models use the same emissions, differences in model results can be ascribed to differences in transport (advection, including also convection and diffusion) only. Secondly we investigate CO as a reactive component of the atmosphere, participating in chemical reactions. In addition to direct sources, CO is also formed by oxidation of NMVOCs, and to a minor extent removed by dry deposition. The main sink for CO is the reaction with OH, and thus differences in OH is one of the main factors affecting CO. Finally we look at ozone. The causes of the differences in calculated ozone are hard to identify due to the highly non-linear couplings of ozone production and destruction, but some clues can be identified based on the calculations of the CO like tracer and CO.

In this paper we first briefly discuss the model comparison to measurements in section 3. In section 4 we go on to describe the source receptor relationships for Europe, including a discussion on how and why the model results differ. Finally, in section 5 we sum up the results for the individual models. Based on model performance compared to measurements and where and when deviations in model results compared to the other models occur we try to indicate the origins of the differences in model behaviour. In the conclusions we then suggest some directions on how this information could be used to harmonize and improve future model calculations.

## 2  The HTAP2 model setup

The HTAP2 model experiment was set up by the Task Force on Hemispheric Transport of Air Pollution (TF HTAP). A project work plan, a description of the model experiments etc. can be found on the TF HTAP web page (http://www.htap.org/). The models were required to perform a 6 month spin-up for all model runs. A more detailed description of the requested model runs, emissions, requested model output and formats etc. is also included in (Galmarini et al., 2017) and references therein. A detailed description of the emissions can be found in Janssens-Maenhout et al. (2015). More documentation about the models can also be found in the supplementary material.

In this paper we focus on the effects on Europe. Even though a substantial number of models have uploaded their results to the database, model results relevant to this publication for ozone (and CO) are only available from 7 of the models for the BASE model runs and for at least the two scenario runs reducing all anthropogenic emissions except $CH_4$ by 20% globally (GLOALL) and in Europe (EURALL). These models have different resolutions, advection schemes, chemical mechanisms etc (see supplementary material and references therein). Additional model runs reducing all anthropogenic emissions in North America (NAMALL), East Asia (EASALL), South Asia (SASALL), Middle East (MDEALL), Russia, Belarus, Ukraine (RBUALL) and ship emissions (OCNALL) are also discussed here. The definitions of these regions are given in Koffi et al. (2016). The models are a subset of the HTAP2 models listed and described in Stjern et al. (2016). Since then additional model result have also been provided for the GFDL_AM3 model, raising the number of models to 8. (GFDL_AM3 model data are included in the database, but in a different format than the other models). Additional information on the models are also listed in the supplementary material. Access to model data are available upon registration from http://aerocom.met.no.

## 3 Models vs measurements

In this section we discuss the performance of the models compared to measurements. Wherever possible we have used the validation tools provided online by AEROCOM: http://aerocom.met.no/ cgi-bin/aerocom/surfobs_annualrs.pl?PROJECT=HTAP&MODELLIST=HTAP-phaseII. This enables the reader to explore the results on their own. For ozone a comprehensive model to measurement comparison is published in Galmarini et al. (2018), including a comparison of both global and regional model results. However, this study focus mainly on the ensemble mean, and individual model results are treated anonymously. For surface ozone we refer to this paper, but additional model validation is also included here. Comparisons of model calculated vertical profiles to ozone soundings are included in the supplementary material. As the focus of this paper is on Europe, only European sites are shown. We have only included models with model output also for the GLOALL and the EURALL scenarios.

### 3.1 Surface

Monthly averaged timeseries of measured versus model calculated CO are shown in the supplementary material for a number of European GAW (Global Atmospheric Watch) sites. Some statistics for these sites are listed in Table 1. At most sites modelled and measured CO has a clear winter maximum and a summer minimum. All models in general reproduce the seasonal cycle well at most sites (see supplementary material), also reflected in their high correlations with the measurements. Correlations shown here are in the same range as correlations with MOPITT satellite measurements as reported by Naik et al. (2013). However, as shown in Table 5, all models except IFS_v2 underestimate annual CO levels by 13% or more. Similar underestimations was also shown in Strode et al. (2015).

The results for the two CHASER model versions with high (1.1 × 1.1 degrees) versus low (2.8 × 2.8 degrees) resolutions differ, but they are qualitatively similar.

This study also includes an evaluation of model results at several mountain sites. Results for these sites are shown, but should be interpreted with caution. The elevations of mountain sites are poorly resolved in the models. Furthermore concentrations are likely to be affected by sub scale circulation patterns as mountain subsidence and upslope winds etc, that are not resolved by the models.

A more comprehensive comparison of the Base model calculations and ozone measurements from the EMEP and airbase measurement networks is presented in Galmarini et al. (2018) as part of HTAP2 and AQMEII (Air Quality Modelling Evaluation International Initiative). However, in the Galmarini et al. (2018) study the main focus is on the ensemble mean. An additional model validation of surface ozone is therefore also included here. Monthly averaged timeseries of measured versus model calculated $O_3$ are shown in the supplementary material for a number of European GAW sites. Some statistics for these sites are listed in Table 2. The GAW sites are background sites relatively far from major sources. Scatter plots for the BASE model runs for ozone versus measurements are shown in the supplementary material. A summary of these results are also presented in Table 5.

With coarse resolution, global models can not be expected to fully reproduce the measurements. The effects on model resolution on the validation of ozone measurements is demonstrated in Schaap et al. (2015) running the same set of models with variable horizontal resolutions. They show that for sites affected by local sources ozone is often over-predicted with coarse resolution as titration effects are watered out. Thus one may expect coarse global models to over-predict ozone levels at several sites classified as background sites. As shown in the scatter plots only the OsloCTM3_v2 and the IFS_v2 model under-predicts the European annual ozone measurements by 22 and 18 percent, the other models overestimate ozone levels by 10 - 22%. This pattern of over and under-estimation is also apparent when comparing the individual GAW sites.

## 3.2 Vertical ozone profiles

Seasonal model calculated vertical profiles of ozone are compared to ozone sonde measurements downloaded from the World Ozone and Ultraviolet Radiation Data Centre (https://woudc.org/home.phpa) for several European sites in the supplementary material. Model calculated profiles are included in the calculations for the approximate same point in time (to the nearest hour) as the ozone sondes, and then averaged seasonally. The number of soundings included in the average for any site and season is listed in the individual panels. The figures have been produced by the AEROCOM tool: http://aerocom.met.no/cgi-bin/aerocom/surfobs_annualrs.pl?PROJECT=HTAP&MODELLIST=HTAP-phaseII.

The profile comparison allows to identify differences between the models in vertical mixing of ozone useful for further interpretation in inter-hemispheric transport efficiency. Note that the GEOS-Chem model only simulates ozone in the troposphere and its ozone levels above 300 hPa should be disregarded. With a relatively inactive chemistry in the winter months the measured ozone profiles

at these sites show little vertical variability, with ozone mixing ratios in the troposphere increasing gradually with height. Model calculated ozone profiles are in general close to the measurements. As the chemical activity increases in Spring and summer months the vertical variability increases, reflecting air masses of significantly different photochemical history at different levels. As was shown in Jonson et al. (2010) the models are not capable of reproducing this vertical structure in ozone levels. Most of the models underestimate free tropospheric ozone in the summer months.

## 4   Source attribution, focusing on Europe

In this section we use the models to attribute the sources of ozone from different world regions, focusing on effects on European ozone levels. In order to better understand the differences between the models, we use a step-wise approach, starting the discussion with the CO like tracer in section 4.1, then we compare results for CO in section 4.2, where the treatment of the sources should be similar in all models, and the main sink is through the reaction with OH. Finally, in section 4.3 we compare the model results for $O_3$.

The calculations of the anthropogenic contributions from the different source regions are based on the difference between the base model runs and HTAP2 model scenario runs reducing all anthropogenic emissions globally (GLOALL), in addition to the reductions in the specific HTAP2 regions. We first compare the model calculated effects of the GLOALL scenario for vertical trans-sections, and discuss the source allocation of domestic European anthropogenic sources versus external transcontinental anthropogenic sources expressed as RERER (Response to Extra-Regional Emission Reductions) as defined in Galmarini et al. (2017):

$$RERER = \frac{EURALL - GLOALL}{BASE - GLOALL}.$$

Again, BASE is the reference model run and EURALL the model runs reducing all European anthropogenic emissions by 20%. RERER is then a measure of the effects of external trans-continental versus domestic European emissions on the species in question. Assuming a fully linear chemistry, a RERER of one means that the concentrations in Europe are completely determined by sources outside Europe, whereas a RERER of 0 means that concentrations are determined by European sources alone. Unfortunately the chemistry is often far from linear. In particular for ozone, ozone titration, mainly in the winter months, can result in RERER values well above one, and in some cases even negative. In the section below annual RERER values are given for Europe as a whole and for four separate receptor regions, NW, SW, SE and GR+TU as shown in Figure 1.

For ozone we also show the source attribution of European ozone further split into separate world regions for the the different models on a seasonal basis in subsection 4.4. Finally in subsection 4.5 we discuss to what extent the choice of ozone metrics will affect our findings.

### 4.1 CO tracer

The CO tracer is calculated with the same anthropogenic emissions as CO, and with a fixed rate of decay giving a lifetime of 50 days. Any differences between the individual models can then be attributed to differences in transport processes. RERER for the CO tracers should be linear as there is no chemical interaction.

Table 3, lists RERER calculated by the EMEP_rv48 and the IFS_v2 models (unfortunately the GFDL_AM3 model only reported BASE and GLOALL and not EURALL for the CO tracer so RERER could not be caclulated) for Europe and the four European sub regions. For Europe as a whole, RERER is also shown in Figure 2. For the CO tracer RERER is ranging from 0.35 to 0.60, depending on model and European sub-region. There is a moderate difference in RERER between the two models. The highest RERER is calculated for the Gr+Tr region as this region is close to regions outside Europe as Russia, Belarus, the Ukraine, the Middle East and also the Mediterranean Sea and Black Sea.

Figure 3a,d,g shows the annual mean difference in BASE - GLOALL of longitudinal CO tracer concentrations as an average between 30 and 60 degrees north. For all 3 models (EMEP_rv48, IFS_v2 and GFDL_AM3) the largest impacts of the 20% emission reduction on concentrations can be seen over the source continents in North America, Europe and in particular over East Asia. There are marked differences between the models as to what extent the CO tracer from the polluted boundary layer is lifted into the free troposphere. The EMEP_rv48 model (Figure 3b), with high RERER (Table 3) has higher tracer contributions in the free troposphere than the other two models (Figure 3d,g). For the tracer the single factor that affects the concentrations is advection. Thus, the differences in the results are caused by various degrees of lifting into the free troposphere, possibly through strong convection, followed by rapid transport further from its sources, subsequently contributing more to the tracer levels in distant regions before being decayed.

The seasonal cycle of the difference in BASE - GLOALL the over Europe, defined as the area bounded by 10°W to 35°E and 30 to 60 °N, is shown in Figure 4a,d,g. This area roughly corresponds to the European regions as shown in Figure 1, but also some additional land and sea areas. The main focus of the figure is in the free troposphere where horizontal gradients in concentrations are small. Liu et al. (2009) calculated the correlations between nearby pairs of sonde stations. They found low correlations near the surface indicating that local and regional effects are important here. From the surface correlations rose sharply to a local maximum in the lower troposphere. We therefore conclude that the selected area is a good representation of the atmosphere above Europe.

There are moderate differences in the seasonal behaviour of the CO tracer between the models, but tracer levels in the free troposphere are again highest in the EMEP_rv48 model. Differences in mixing ratios peak in the first part of the year when emissions are high and the exchange between the boundary layer and the free troposphere over Europe is weak. Differences in the free troposphere may reflect CO tracer advected from regions upwind with convective activity also in winter, or in

the preceding autumn months increasing the free tropospheric reservoir in the following winter and spring.

## 4.2 CO

Emissions of CO and the CO tracer are identical, and the vertical and seasonal extent of the CO perturbation resembles the results for the CO tracer in section 4.1. CO also has a number of natural
sources and is also produced by oxidation of $CH_4$ and NMVOC. But these are not very relevant for the perturbation results. The dominant sink for CO in the atmosphere is the reaction with the OH radical, with a winter minimum and peaking in summer.

Table 3 lists RERER values for the seven models for Europe as a whole and for the four European sub regions shown in Figure 1. RERER is ranging from 0.24 to 0.71, depending on model and
European sub-region. Differences between the models are now caused by transport (as for the CO tracer) and chemistry. The difference in RERER between the EMEP_rv48 and IFS_v2 models is slightly larger for CO than for the CO tracer. Assuming that the CO chemistry is close to linear, this indicates a longer lifetime in the atmosphere than the 50 days for the CO tracer. IPCC Working group 1: the scientific basis (IPCC WG1, 2001), https://www.ipcc.ch/ipccreports/tar/wg1/130.htm#tab41a)
reports a lifetime of 0.08 to 0.25 years (about 30 to 90 days) depending on location and season, on average longer than 50 days.

As shown in Table 3 and Figure 2, the spread in RERER between the models is again moderate. For the EMEP_rv48 and IFS_v2 models the difference in RERER is slightly larger than for the CO tracer. As for the CO tracer, the highest RERER is in general calculated for the GR+TR region as this
region is close to the outer border of the European domain.

Figure 3b,e,h,k,m,o,q shows the annual mean difference in BASE - GLOALL CO concentrations as an average between 30 and 60 degrees north. For all the models large differences in concentrations can be seen over the polluted continents North America, Europe and in particular over East Asia. As for RERER, there are differences between the models, in particular in the free troposphere.
The EMEP_rv48 model (Figure 3b), with high RERER, has higher CO contributions in the free troposphere than the other models. For the other two models (IFS_v2 and GFDL_AM3) the results for CO and the CO tracer are more similar, indicating a chemical lifetime closer to 50 days while in the EMEP_rv48 model CO seems to have a longer lifetime.

As CO is lifted into the free troposphere transport between continents is rapid, and CO can be
transported further before decaying. This suggests that as for the CO tracer RERER to a large extent is controlled by the level of rapid lifting and subsequent efficient intercontinental transport in the free troposphere.

The seasonal cycle of the difference in BASE - GLOALL over Europe is shown in Figure 4, middle panels. As for the CO tracer, differences in concentrations peak near the surface in the first part of the

year when emissions are high and the exchange between the boundary layer and the free troposphere is weak. In addition the differences are magnified by the seasonal cycle in the OH sink.

We don't have access to the OH levels for all the models, but for those models providing OH (EMEP_rv4.8, CHASER_re1, OsloCTM3_v2 and CAMchem) annually averaged tropospheric levels are shown in the supplementary material along with the difference between the average and the four

individual models. OH levels in the EMEP_rv4.8 model are low compared to the average, at least in the upper and middle troposphere. This may lead us to suspect that the widening gap in RERER from CO tracer to CO between the IFS_v2 and the EMEP_rv4.8 model is caused by differences in OH (however, this can not be confirmed, as OH is not available from the IFS_v2 model). Likewise, the higher than average OH levels in the OsloCTM3_v2 model may explain the lower than average CO

RERER values for this model.

Furthermore the lifting of pollutants from the boundary level to the free troposphere is likely to affect the chemistry in the free troposphere causing parts of the differences in OH. The EMEP_rv48 model does not perturb aircraft emissions in the BASE-GLOALL scenario, and this could explain some of the differences between this model and the 3 other models. See also discussion on ozone in

section 4.3 below.

### 4.3  $O_3$

Tropospheric ozone differs from CO and the CO tracer as it is not emitted, but rather it is a secondary product involving combinations of chemical production and loss processes, exchange with the stratosphere, surface deposition and transport. Ozone in the troposphere is advected from the

stratosphere mainly by stratospheric folding events, but its main sources (and sinks) are in the troposphere (TF HTAP, 2010; Stevenson et al., 2006). Net ozone production require ample sunlight and a sufficient supply (and mix) of mainly NMVOC (Non-Methane Volatile Organic Compounds), $CH_4$ CO and $NO_x$.

Table 3, lists annual average RERER, for Europe and for the four European sub regions. RERER is

ranging from 0.56 to 1.38, depending on model and European sub-region. As seen in Table 3 and Figure 2 $O_3$ RERER values are higher than for the CO tracer and for CO. Lifetimes for ozone in the troposphere is highly variable, depending on season and altitude, ranging from hours to a few days in the boundary layer to weeks and even months in the free troposphere (TF HTAP, 2010). However, the overall lifetime in the troposphere is shorter than for CO, see also IPCC Working group 1: the

scientific basis (IPCC WG1, 2001), Table 4.1a. The high RERER values are therefore caused by the non-linear chemistry that for some models can result in RERER values even exceeding one, and for seasonal RERER even negative values (not shown). The spread in RERER between the individual models is markedly larger than for CO and the CO tracer. Differences in transport, depositions and in particular a nonlinear chemistry, give substantial room for variability in ozone levels between the

models. In NW Europe low amounts of UV radiation, inhibiting rapid photochemistry throughout

much of the year, as a result of its northerly location and high cloud fractions, in combination with high $NO_x$ emissions, result in ozone titration and calculated RERER around 1 for a majority of the models. The lowest RERER is calculated for the Gr+Tr (Greece + Turkey) and partially SW European regions. The EMEP_rv48 and the IFS_v2 are the only two models where RERER can be calculated for the CO tracer, CO and ozone. Whereas for the CO tracer and CO IFS_v2 RERER is close to 0.5, it jumps to well above one for ozone, well above any ozone RERER value found by the other models. To a lesser extent this also applies to the CAMChem and GEOS-Chem models. Even though the GEOS-Chem and the OsloCTM3_v2 models have the lowest RERER for CO, the ozone RERER is well above the ensemble mean. The CHASER models are close to the ensemble mean for CO, but has the lowest RERER for ozone. The EMEP model has the highest RERER for CO and the CO tracer, but is close to the ensemble mean for ozone. These changes in positions between CO and ozone are likely caused by differences in the combined interactions of transport and chemistry.

Based on the HTAP2 model calculations, Huang et al. (2017) have calculated RERER for the North American continent. In general these RERER values are markedly lower than those found here for Europe, indicating a larger amount of ozone produced locally over the North American source region. Located further north, Europe is receiving less UV radiation than North America. Europe is also affected by nearby source regions such as Russia, Belarus, Ukraine, the Middle East, North Africa and shipping. These two factors likely explain the higher RERER values over Europe compared to North America.

Figure 3c,f,i and 4d,e,f shows the annual longitudinal mean difference in BASE - GLOALL $O_3$ concentrations as an average between 30 and 60 degrees north. The differences between the models are again markedly larger than for CO and the CO tracer. One notable difference stems from the interpretation of the scenario definition. The OsloCTM3_v2 model, CAMchem model and the CHASER models have included a 20% emission reduction also in aircraft emissions in the GLOALL scenario, whereas the EMEP_rv48 model, the IFS_v2, the GFDL_AM3 and the GEOS-Chem models have not. As a result the additional ozone from BASE - GLOALL is much higher in the middle and upper troposphere by ca. 2 ppb at 300 hPa (10 km), and 3 ppb at 200 hPa (12 km) for the first three models listed. For the OsloCTM3_v2 model the $O_3$ signal from aircraft emissions is located much lower in the troposphere than for the CAMchem and CHASER models. $O_3$ in the lower troposphere, and in particular in the boundary layer, appears to be not so much affected by aircraft emissions. Based on several global models, run with and without aircraft emissions with aircraft emissions of similar magnitude as used in this study Figure 7 in Cameron et al. (2017) suggests a median zonal perturbation of 1 ppb (range 0 to 3 ppb) at 300 hPa, and 1.7 ppb ( range 0 to 8 ppb) at 200 hPa, scaling their results to a 20% emission perturbation. These results are very consistent with our finding for the aircraft perturbation at these altitudes.

As is the case for CO and the CO tracer, the EMEP_rv48 model (Figure 3c), has higher $O_3$ contributions in the free troposphere than the IFS_v2, GFDL_AM3 and GEOS-Chem models (the

three other models not perturbing aircraft emissions). This could be caused by lifting of ozone and ozone precursors from the boundary layer into the free troposphere and subsequent rapid transport

between continents in the free troposphere.

The seasonal cycle of the difference in BASE - GLOALL over Europe is shown in Figure 4 right panels. Whereas the contributions from aircraft peaks in summer and autumn, the differences in BASE - GLOALL in general peaks in spring in the lower troposphere except for the CAMchem and GFDL_AM3 models peaking in mid summer. The CAMchem model has very high European net

surface ozone contribution in summer compared to contributions from other regions, contributing to the shift in the seasonal maximum from spring into summer. See also discussion in sections 4.4 and 4.5 below.

### 4.4 European $O_3$ source attribution by world region

Based on the difference between the BASE model runs and the 20% perturbations of global and

European emissions we attribute a major portion of ozone of anthropogenic origin in Europe to sources outside Europe. As part of the HTAP2 requests, model calculations have also been made reducing anthropogenic emissions by 20% in other major world regions. In Figure 5 the contributions to European ozone levels calculated by the different models are shown with sources originating from these different world regions. None of the models have made the calculations for all the regions. For

each model the contribution from ROW (Rest Of the World) is calculated by subtracting the sum of the contributions from from available world regions from the BASE - GLOALL contribution. Thus the portion related to ROW includes a varying aggregation of world region definitions depending on the model. In addition the percentage contributions to annual average ozone and summer ozone to Europe (letting the GLOALL perturbation represent 100%) from Europe, North America and East

Asia, based on the numbers shown in Figure 5, are shown in Table 4. The percentage contributions to SOMO35 and $POD_1$ forest is also given in this table (see section 4.5 for definitions of SOMO35 and $POD_1$ forest).

There are large differences between the models, in particular for the contributions of annual ozone from Europe, ranging from -48 to +37 percent. However, the contributions to summer ozone are

much more similar ranging from 35 to 55 percent. Still, there are some common features: For all models and all seasons except for the CHASER_re1 in summer, the contributions from regions outside Europe are larger than the contribution from European sources. The contributions from non European sources are largest in Spring (Figure 5). The largest non European contributions are from North America (NAMALL) and East Asia (EASALL). Contributions from Russia, Belarus,

Ukraine (RBUALL) are mixed, with significant calculated contributions calculated by two models (EMEP_rv48 and CHASER_re1). Contributions from the middle East (MDEALL) and North Africa (NAFALL) are small. There are also substantial contributions from ocean shipping (OCNALL), but this source has only been calculated by the EMEP_rv48 model. For Europe contributions from

shipping has also been shown in other studies as as Jonson et al. (2015) using the EMEP regional

model and Brandt et al. (2013) using a different (non HTAP2) model. For all models, except the CHASER models (represented by CHASER_re1 in Figure 5), ozone titration dominates the overall European contributions when summed up over the three winter months. However, for all the models, including also the CHASER_re1 model, the net European contributions includes regions of net ozone production and net ozone destruction in winter.

The negative, or close to zero, net annual ozone production over Europe in the IFS_v2, GEOS-Chem and CAMChem models can explain the increase in RERER from CO to ozone in Figure 2 discussed in section 4.3. Likewise also the corresponding relative decrease in RERER for the CHASER models, and partially the EMEP_rv48 model can be explained by positive net ozone production over Europe.

In comparison to HTAP1, HTAP2 regions are better confined to the political boundaries on the

continent, and hence more policy relevant. In addition emissions as well as models are up-to-date. To disentangle whether the changes from HTAP1 to HTAP2 are due to emissions, a changed model ensemble or changes in receptor regions is unfortunately not possible in a fully quantitative way. Source and receptor regions have been chosen in HTAP2 to cover the land-only politically connected regions accurately on a 0.1 degree grid. In HTAP1 the EUR region was a simple latitude - longitude

box, also including parts of North Africa, the Middle East, Russia, Belarus, Ukraine and large sea areas, all of these identified as non European regions in HTAP2. In HTAP2 the European region is smaller, thus exporting larger fractions to nearby regions, but most major HTAP1 source regions are located within the smaller HTAP2 region, thus making this region more sensitive to titration effects. As a result the effects of emissions on ozone levels from the EUR region to itself is reduced.

The ensemble mean contribution to annual mean ozone levels from Europe to itself has decreased from 0.82 ±0.29 ppb in HTAP1 to just 0.11 ±0.32 ppb in HTAP2. Also - total and regional distribution of emissions for the base year changed from HTAP1 (2001) to HTAP2 (2010). Gaudel et al. (2018) have analysed the ozone trends between the years 2000 and 2014. Over Europe. They found a general ozone increase in the winter months (December, January, February) and a general decrease in the

summer months (June, July, August). The emission trends in the HTAP1 world regions are given in Turnock et al. (2018) between 2001 (the base year for HTAP1) and 2010. The changes in measured ozone are consistent with the reductions in European (and North American) emissions of $NO_x$ (along with other ozone precursors) over the same period resulting in less titration and thus increased ozone levels in some areas mainly in the winter months, and at the same time less net ozone production

in summer. Likewise emissions in North America have decreased and may explain the 0.37 ±0.10 in HTAP1 to 0.22 ±0.07 ppb (HTAP2) decrease in the ensemble mean contributions from North America to European ozone levels. Over the same period emissions in other world regions as East Asia have increased. This increase may explain the 0.17 ±0.05 to 0.22 ±0.13 ppb ensemble mean increase from HTAP1 to HTAP2 in the East Asian contribution to European ozone levels. Contributions from

South Asia are small in both HTAP1 and HTAP2 (0.07 versus 0.05).

A combined effect of the change in the definition of the European domain and the changes in emissions is that the relative model calculated contributions to surface ozone levels from non European sources is much larger in HTAP2 compared to HTAP1. In the HTAP1 final report (TF HTAP (2010), Table 4.2) the concept of RAIR (Relative Annual Intercontinental Response), defined as the ratio of the response in a particular region (Europe) due to the combined influence of sources in the three other regions (North America, East Asia and South Asia) to the response from all these four source regions. RAIR for the models in Figure 5 is 82% as opposed to 43% in the HTAP1 final report.

Using tagging in a regional model the calculated contributions from non European sources have also been calculated by Karamchandani et al. (2017). They calculate a much smaller contribution from non European sources than in this study, similar to the contributions calculated in HTAP1. In the Karamchandani et al. (2017) study non European ozone is defined as the boundary influx to the model domain. As a result shipping, and nearby non Central European regions, are included in the domain, similar to the definition of the HTAP1 European domain.

### 4.4.1 Effects of a 20% $CH_4$ perturbation

As shown in Figure 5 four of the models have also calculated the effects of a 20% increase in $CH_4$ concentrations. While these concentration perturbations are not directly comparable to air pollutant emission perturbations, they correspond to an uncertainty in $CH_4$ change in 2030 from the $5^{th}$ Coupled Model Intercomparison Project (CMIP5) for the RCP8.5 and RCP2.6 scenarios. Averaged over the four models the calculated effects for Europe of 20% changes in $CH_4$ levels is almost three quarters of the effects of the BASE - GLOALL model runs. However, comparing a 20% change in $CH_4$ concentrations, and the effects of the GLOALL emission scenario requires careful interpretation. Because of its relatively long lifetime of the order of 10 years in the atmosphere, a 20% change in concentration corresponds to an approximate 40 year long historic $CH_4$ trend (Meinshausen et al., 2011). The GLOALL scenario is not accounting for the full impact of a continued 20% reduction in emissions. With a continued emission reduction scenario, the overall ozone reductions would be larger, while the methane attributable fraction, relatively, would be smaller. The effects of $CH_4$ is insensitive to the location of the emissions, and there are only moderate differences in the response in ozone levels by world region (Fiore et al., 2008). The agreement between the model estimates is a lot better for the $CH_4$ perturbation compared to the BASE - GLOALL estimates, and not too different for the HTAP1 estimate of about 1 ppb (Fiore et al., 2008). The sensitivity of ozone to $CH_4$ is discussed in more detail in Turnock et al. (2018).

### 4.5 Does the choice of ozone metric matter?

In Figure 5 the contributions to European ozone levels are shown as seasonal and annually averaged ozone and in Table 4 the percentage contributions to annual and summer ozone from European, North American and East Asian sources are listed based on the numbers from Figure 5. In Europe several

other metrics are also used calculating the effects of ground level ozone. The two metrics listed below are designed to capture the effects of ground level ozone on human health (SOMO35) and on the environment (POD$_1$ forest):

- SOMO35: Sum of Ozone Means Over 35 ppb is the indicator for health impact assessment recommended by WHO. It is defined as the yearly sum of the daily maximum of the running 8-hour running average of ozone above 35 ppb.

- POD$_1$ (deciduous) forest: Phyto-toxic Ozone Dose for forests is the accumulated stomatal ozone flux over a threshold Y integrated from the start to the end of the growing season. For deciduous forests, discussed here, the critical level of 4 mmol m$^{-2}$ is exceeded in most of Europe, indicating a risk of ozone damage to forests. See Mills et al. (2011a, b) for further description of this metric.

POD$_1$ forest is only accumulated over the growing season in summer when the contributions from local European sources are high. Likewise SOMO35, with a cutoff value at 35 ppb, is accumulated mainly in the summer months. Thus both metrics these netrics largely exclude the effects of ozone titration mainly taking place in other seasons.

Contributions to annual mean ozone are accumulated regardless of season and ambient ozone levels. In the EMEP_rv48 model contributions from NAMALL and EASALL have already been shown to be little affected by ozone titration and a major source mainly in the spring months before the local European sources gathers momentum. Contributions from RBUALL and OCNALL are a mixture of nearby and more distant sources, and effects on annual mean ozone, SOMO35 and POD$_1$ forest are similar. In Jonson et al. (2018) it is shown that the anthropogenic percentage contribution to these ozone indicators in Europe are substantially higher than for annually averaged ozone when isolating the contributions from nearby sea areas, similar to the effects of Europe on itself. On the other hand ozone from distant sea areas contributes more outside the summer months. It is likely that the difference between the ozone metrics would be considerably larger if calculated with the other models, and in particular those models with substantial titration effects from European Emissions as already shown in Figure 5.

Unfortunately the two latter metrics have only been provided by the EMEP_rv48 model. The annual effects of the 20% reductions in anthropogenic emissions from different world regions are shown for annual mean ozone, SOMO35 and POD$_1$ forest in Figure 6 as percentage contributions where 100% refers to the difference between the BASE and GLOALL scenario. The regional contributions, expressed by these metrics, are also listed in table 4. The figure and table clearly shows that the choice of metric matters, in particular for the effects of European Emissions. POD$_1$ forest is accumulated in the growing season in summer. A large portion of SOMO35 is also accumulated in the summer months. Table 4 also lists the percentage contributions to summer ozone for all models. The similarities in the

percentages for summer ozone and the ozone metrics in EMEP_rv48 is an indication that also for the other models these percentages are comparable.

## 5   Discussion on individual models

As shown above differences between the models amplify going from the simple CO tracer, via CO, to ozone. This stepwise amplification provides an opportunity to pinpoint probable causes. At the same time we also use the comparisons to measurements as a guidance. Some of the results from the individual model calculations are summed up in Table 5. Below we discuss the characteristics and the results for the individual models. Here we try to point out if, and at what stage, the results from the individual models deviate from the other models. It should be stressed that such a deviation does not necessarily imply that the results from a particular model is wrong.

The horizontal resolution of the EMEP_rv48 model is $0.5 \times 0.5$ degrees, higher than any of the other models. Compared to the other models, the difference between BASE and GLOALL is among the highest compared to the other models for CO and the CO tracer. Much of this may be caused by a larger rate of exchange (possibly by convection) between the boundary layer and the free troposphere as indicated by the CO tracer. On the other hand this model performs among the best both for CO and ozone compared to measurements. Calculated CO levels at remote sites have a small, low bias and are well correlated compared to the other models, see Table 1) and supplementary material. The model is one of the models with highest overestimation of ozone in the free troposphere in the winter and spring months.

The horizontal resolution of the IFS_v2 model is $0.7 \times 0.7$ degrees. The RERER results for CO are close to the ensemble mean and CO levels close to observations. For ozone RERER is higher than the other models, and above 1 in all European regions except Greece and Turkey. European net Ozone production is strongly affected by ozone titration resulting in net ozone loss from European sources in all seasons except summer. Calculated ozone levels in Europe are low compared to measurements, in particular for low ozone sites. The IFS_v2 model differs from the other models by having the highest level of ozone titration. The underestimation of ozone at low ozone sites is most likely caused by the high level of titration.

The horizontal resolution of the OsloCTM3_v2 model is $2.8 \times 2.8$ degrees. The advection is solved using the Prather scheme, giving very little numerical diffusion. For CO RERER is well below the model ensemble mean. The model underestimates CO, and overestimates $O_3$ compared to measurements. For CO the low RERER and the underestimation of surface CO compared to measurements could be affected by higher OH values compared to the other models.

The two models CHASER_re1 (resolution $2.8 \times 2.8$ degrees) and CHASER_t106 (resolution $1.1 \times 1.1$ degrees) differ only in resolution, and results from the two models are very similar. RERER for CO is close to ensemble mean. RERER for ozone almost 30% lower than ensemble mean. The

CHASER models differs from the other models by having lower RERER for ozone and little or no ozone titration over Europe even in winter. The lack of ozone titration may be the cause of the overestimation of ozone at low ozone sites seen in the ozone scatter plot shown in the supplement.

The horizontal resolution of the GEOS-Chem model is $2.0 \times 2.5$ degrees. CO concentrations on acerage underestimated by more than 20 percent. $O_3$ concentrations overestimated by 14%. $O_3$ is only simulated in the troposphere and ozone levels above the tropopause are based on boundary concentrations (see supplementary material) and should be disregarded here. Like most models the GEOS-Chem model underestimates CO and overestimates $O_3$ in EU. The GEOS-Chem model has the lowest RERER value for CO, but at the same time a high RERER for ozone. It has high ozone titration in winter and high European ozone production in summer. As for the IFS_v2 model the underestimation of ozone at low ozone sites is most likely caused by the high level of titration.

RERER calculated by the GFDL_AM3 model is close to the ensemble mean for both CO and $O_3$. RERER for CO 20% below ensemble mean. RERER o3 17% higher than the ensemble mean.

The horizontal resolution of the CAMchem model is $1.9 \times 2.5$ degrees. CO concentrations are on average underestimated by 25% and $O_3$ concentrations are overestimated by 22%. RERER is close to ensemble mean for both CO and $O_3$. Similar to the GEOS-Chem model the CAMchem model has high RERER for ozone in combination with high ozone titration in winter and high European ozone production in summer. The high net ozone production in summer is the likely cause for the shift in the $O_3$ maximum for BASE - GLOALL from Spring to Summer in the lower troposhere above Europe.

## 6  Conclusions

The HTAP1 experiment showed a very large spread in model results. (TF HTAP, 2010). Part of this spread may have been caused by differences in the 2001 emissions, as each modelling group used their own set of emissions. In HTAP2 all models are required to use a common set of emissions. Even so, the spread in model results remains large. The model calculated relative contributions to annual surface ozone levels from non European sources is much larger in HTAP2 compared to HTAP1. The main reason for this is that the contributions from Europe to itself has decreased from 0.82 $\pm$0.29 ppb to just 0.11 $\pm$0.32 ppb. At the same time calculated contributions from North America have decreased far less, from 0.37 $\pm$0.10 ppb to 0.22 $\pm$0.07 ppb, and increased from East Asia from 0.17 $\pm$0.05 ppb to 0.22 $\pm$0.13 ppb. As a result RAIR (the metric used in HTAP1) has increased from 43 to 82%. In parts this difference could be explained by decreasing emissions in Europe and increased emissions in most other regions as East Asia from year 2001 to 2010. However, the results from the two HTAP phases can not easily be compared, partially because the model ensemble has changed, but mainly because the definition of the European area has changed considerably from HTAP1 to HTAP2. In HTAP2 the contributions to anthropogenic ozone from in particular ocean shipping and from nearby Russia, Belarus and Ukraine are of the order of 10%. Parts of these regions were included

as European and thus also contributed to the higher RAIR in HTAP1. The HTAP2 source and receptor regions are now better designed for characterising export and import of air pollution to and from the individual regions.

Calculations with the EMEP_rv4.8 model indicate that the contributions to European annual average ozone and ozone indicators of anthropogenic origin from shipping are all of the order 10%.

For HTAP2 additional diagnostics were defined which allow better understanding of transport efficiencies, such as the utilisation of idealized CO tracer and more information on the vertical distribution of tracers in the output requirements.

Not surprisingly, our study reveals that the magnitude of the inter-model spread in hemispheric transport, characterised by RERER, increases with the complexity of the processes involved. We demonstrate that the spread in European RERER increases from the idealised CO tracer to fully prognostic CO and ozone. Atmospheric transport alone can not be made responsible for the larger spread between the models in RERER going from CO to ozone. As the residence time in the troposphere is longer for CO compared to ozone (see discussion in sections 4.2 and 4.3). the increase in RERER from CO to $O_3$ must be caused by more complex non-linear chemistry forming and destroying ozone and not by a longer atmospheric lifetime of $O_3$ compared to CO.

The model resolution differs between the individual models. Model results from the two CHASER models, differing in model resolution only, are quantitatively similar when compared to measured CO and $O_3$ at background measurement sites and very similar in RERER for CO and $O_3$, suggesting that resolution differences at the scales investigated here, are not important to explain RERER differences between the global models. Still, it is difficult to conclude in general to what extent horizontal resolution affects the source receptor calculations at intercontinental scales.

The joint and consistent analysis of a CO tracer, CO and $O_3$ in this paper is a tool in understanding where and why (right or wrong) the models differ, however, it has a potential for wider use, enhancing our understanding of the result and also as a tool for model improvements, reducing the overall uncertainty in future model calculations. We believe that in order to close the gap in model results, and subsequently improving the reliability of the model output, possible future model inter-comparisons should be more process oriented (transport, depositions, chemistry etc). Our study shows that models differ already for CO and the inert CO tracer, where differences were established with 2 models, but that differences are amplified as more chemistry is added. Note that the CO RERER and $O_3$ RERER values are not correlated taken the models as samples. The large additional spread in model results for ozone compared to CO and the CO tracer is clearly induced by differences in model chemistry exemplified by the treatment of titration in the winter boundary layer. However, differences in chemistry may well also be induced by differences in advection/convection as the level of exchange will inevitably affect the chemical regime in both the free troposphere and in the boundary layer. We therefore believe that further process oriented evaluations (comparing advection/convection, chemistry,

dry and wet deposition etc separately) should be made, making use of relevant meteorological and chemical measurements.

The HTAP2 results, using state of the art global models, reflecting updated emission estimates and refined receptor region definitions, confirm the importance of hemispheric transport of air pollution. Based on seasonal and annual averaged ozone, all the models agree that the contribution from non European sources to European surface ozone levels is considerable. However, calculations with the EMEP_rv4.8 model shows that this conclusion to some extent will depend on the choice of ozone metrics. Alternative metrics, such as SOMO35 and $POD_1$ forest, will to a larger extent accumulate in

the summer months when ozone production peaks over the European continent. The dependence on ozone metrics seen in the EMEP_rv4.8 model is corroborated by the other HTAP2 models all showing the effects of summer ozone pointing in the same direction. As a result the potential for reducing the detrimental effects from ozone caused by European emissions alone is higher when applying these metrics.

The model results suggest that sizeable reductions in European ozone levels can best be achieved through a combined global effort (or at least throughout the northern hemisphere) to reduce the emissions of ozone precursors. Efforts to curb regional pollution in other non European regions, exemplified by the reductions in North American emissions of ozone precursors, have most likely reduced the ozone burden also in Europe. Further reductions in the Emissions of ozone precursors are

expected in Europe and North America. However, decreases here has so far been partially counteracted by increases elsewhere. Other regions, such as East Asia, are currently facing severe air pollution problems. Part of the remedy for the elevated European ozone levels may well be local and regional air pollution control to curb air pollution also in these regions.

*Acknowledgements.* This work has been partially funded by EMEP under UNECE. Computer time for EMEP

model runs was supported by the Research Council of Norway through the NOTUR project EMEP (NN2890K) for CPU, and NorStore project European Monitoring and Evaluation Programme (NS9005K) for storage of data. The AeroCom database at Met Norway received support from the CLRTAP under the EMEP programme, through the service contract to the European commission no. 07.0307/2011/605671/SER/C3, and benefited from the Research Council of Norway projects no. 229796 (AeroCom-P3) and no. 235548 (SLCF). The National

Center for Atmospheric Research is funded by the National Science foundation. The University of Colorado has received support from NASA under grant NNX16AQ26G. We would also like to thank WOUDC for making the ozonesonde measurements available. Some data used in this publication were obtained as part of the Network for the Detection of Atmospheric Composition Change (NDACC) and are publicly available through http://www.ndacc.org. EMEP surface measurements have been made available through the EBAS web site,

http://ebas.nilu.no/Default.aspx.

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

**Table 1.** Annual mean measured and model calculated CO in ppb for the European CO GAW sites downloaded from http://ds.data.jma.go.jp/gmd/wdcgg/. See also auxiliary material for figures. The comparison is based on monthly average model and measured data. Model IFS2 is IFS_v2, EMEP is EMEP_rv48, GEOS is GEOS-Chem, CAMC is CAMchem, OSLO is OsloCTM3_v2, GFDL is GFDL_AM3 and CHAS are the CHASER models (CHASER_t106/CHASER_re1). **Bold face**/*italic* numbers represent the model calculated concentration with highest/lowest model bias/correlation at the individual sites.

| Site: | Obs. | Calculated concentrations | | | | | | | Correlations | | | | | | |
|---|---|---|---|---|---|---|---|---|---|---|---|---|---|---|---|
| | | IFS2 | EMEP | GEOS | CAMC | OSLO | GFDL | CHAS | IFS2 | EMEP | GEOS | CAMc | OSLO | GFDL | CHAS |
| *Mountain sites* | | | | | | | | | | | | | | | |
| Summit | 121 | 103 | **109** | 87 | 85 | *75* | 84 | 87/88 | 0.92 | *0.89* | 0.94 | 0.91 | 0.93 | 0.91 | 0.91/**0.96** |
| Zugspitze | 153 | 172 | 133 | **146** | 134 | 168 | *130* | 133/130 | 0.61 | 0.57 | 0.62 | 0.45 | **0.65** | *0.25* | 0.57/0.51 |
| Hohenpeiss. | 176 | 200 | 151 | 146 | 134 | **168** | *130* | 133/137 | **0.96** | **0.96** | 0.95 | 0.96 | 0.86 | *0.83* | 0.97/**0.99** |
| Jungfraujoch | 131 | 168 | 141 | 135 | 124 | *185* | **130** | 124/138 | 0.65 | **0.90** | 0.65 | 0.69 | *0.33* | 0.70 | 0.74/0.73 |
| Rigi | 181 | *242* | 138 | 135 | 124 | **185** | 130 | 126/138 | 0.76 | **0.95** | 0.87 | 0.94 | *0.64* | 0.88 | 0.86/0.93 |
| *West and central Europe* | | | | | | | | | | | | | | | |
| Heimaey | 123 | **118** | 108 | 90 | 88 | *77* | 84 | 86/89 | *0.41* | **0.95** | 0.92 | 0.82 | 0.92 | 0.88 | 0.88/0.94 |
| Mace Head | 120 | 109 | **110** | 93 | 90 | *78* | 88 | 91/92 | 0.90 | **0.96** | 0.88 | 0.87 | 0.92 | *0.83* | *0.83*/0.89 |
| Kollumerward | 193 | 158 | 137 | 123 | 118 | **172** | *111* | 131/115 | **0.96** | 0.86 | 0.94 | 0.90 | *0.65* | 0.94 | 0.93/0.89 |
| Neuglobsow | 184 | **151** | 136 | 127 | *118* | 127 | 121 | 127/*118* | **0.98** | *0.81* | 0.96 | 0.91 | 0.88 | 0.95 | 0.88/0.82 |
| Ochsenkopf | 147 | *164* | 142 | **150** | 133 | 131 | 134 | 144/137 | 0.53 | **0.78** | *0.43* | 0.47 | 0.58 | 0.45 | 0.66/0.62 |
| Payern | 216 | **179** | 149 | 135 | *124* | 131 | 130 | 127/127 | 0.91 | 0.85 | **0.96** | 0.81 | 0.78 | *0.61* | 0.90/0.83 |
| Schauinsland | 157 | *212* | **156** | 147 | 136 | 152 | 152 | 142/153 | 0.77 | **0.96** | 0.83 | 0.88 | 0.80 | *0.75* | 0.93/0.89 |
| *Northern Europe* | | | | | | | | | | | | | | | |
| Pallas | 131 | 111 | **114** | 99 | 94 | *78* | 86 | 95/87 | 0.93 | 0.91 | 0.94 | 0.89 | 0.95 | *0.80* | 0.92/**0.96** |
| Zeppelinfjell | 125 | 104 | **111** | 91 | 88 | *77* | 86 | 84/86 | **0.94** | *0.87* | 0.93 | *0.87* | **0.94** | 0.93 | 0.90/**0.94** |
| *South and Eastern Europe* | | | | | | | | | | | | | | | |
| Hegyhatsal | 212 | **164** | 141 | 132 | 126 | *120* | 138 | 134/123 | **0.91** | 0.72 | 0.88 | 0.73 | 0.77 | 0.79 | 0.85/*0.71* |
| Krvavec | 153 | *218* | **148** | 139 | 138 | 125 | 135 | 138/120 | 0.88 | **0.96** | 0.85 | 0.82 | 0.93 | *0.80* | 0.92/0.94 |
| Lampedusa | 128 | **112** | 108 | 95 | 104 | 93 | *91* | 101/101 | 0.82 | **0.94** | 0.86 | *0.53* | 0.68 | 0.66 | 0.85/0.91 |
| Izana | 104 | 95 | **96** | 80 | 79 | *75* | 79 | 85/85 | 0.89 | **0.98** | 0.91 | 0.90 | 0.77 | 0.84 | *0.71*/0.83 |

**Table 2.** Annual mean measured and model calculated O$_3$ in ppb for the European O$_3$ GAW sites downloaded from http://ds.data.jma.go.jp/gmd/wdcgg/. See also auxiliary material for figures. The comparison is based on monthly average model and measured data. Model IFS2 is IFS_v2, EMEP is EMEP_rv48, GEOS is GEOS-Chem, CAMC is CAMchem, OSLO is OsloCTM3_v2, GFDL is GFDL_AM3 and CHAS are the CHASER models (CHASER_t106/CHASER_re1). **Bold face**/*italic* numbers represent the model calculated concentration highest/lowest model bias/correlation.

| Site: | Obs. | Calculated concentrations | | | | | | | Correlations | | | | | | |
|---|---|---|---|---|---|---|---|---|---|---|---|---|---|---|---|
| | | IFS2 | EMEP | GEOS | CAMC | OSLO | GFDL | CHAS | IFS2 | EMEP | GEOS | CAMc | OSLO | GFDL | CHAS |
| Atlantic and northern Europe | | | | | | | | | | | | | | | |
| Summit | 48 | 41 | 44 | 46 | 41 | 29 | 55 | 43 | 0.80 | 0.93 | 0.81 | 0.73 | 0.67 | 0.98 | 0.89 |
| Heimaey | 39 | 27 | 38 | 37 | 32 | 35 | 45 | 32 | 0.84 | 0.93 | 0.94 | 0.98 | 0.85 | 0.85 | 0.99 |
| Mace Head | 36 | 31 | 38 | 38 | 33 | 37 | 43 | 39 | 0.54 | 0.94 | 0.85 | 0.92 | 0.89 | 0.95 | 0.88 |
| Vindeln | 28 | 26 | 31 | 34 | 29 | 25 | 39 | 28 | 0.54 | 0.94 | 0.85 | 0.92 | 0.89 | 0.95 | 0.88 |
| Dobele | 48 | 24 | 34 | 33 | 29 | 24 | 37 | 32 | 0.56 | 0.87 | 0.61 | 0.60 | 0.67 | 0.85 | 0.74 |
| Zoseni | 53 | 24 | 33 | 33 | 28 | 22 | 37 | 32 | -0.11 | 0.43 | -0.04 | -0.05 | 0.66 | 0.62 | 0.13 |
| Rucava | 28 | 27 | 35 | 37 | 32 | 20 | 37 | 30 | 0.68 | 0.94 | 0.54 | 0.60 | 0.82 | 0.53 | 0.86 |
| Central Europe | | | | | | | | | | | | | | | |
| Kollumerwaard | 27 | 22 | 34 | 35 | 31 | 14 | 37 | 33 | 0.84 | 0.95 | 0.71 | 0.79 | 0.68 | 0.85 | 0.94 |
| Waldhof | 28 | 24 | 33 | 29 | 28 | 17 | 34 | 33 | 0.84 | 0.88 | 0.85 | 0.90 | 0.85 | 0.89 | 0.92 |
| Neuglobsow | 28 | 24 | 34 | 33 | 30 | 17 | 37 | 33 | 0.75 | 0.86 | 0.73 | 0.79 | 0.80 | 0.83 | 0.83 |
| Schauinsland | 44 | 22 | 36 | 34 | 32 | 18 | 37 | 40 | 0.91 | 0.76 | 0.94 | 0.94 | 0.88 | 0.92 | 0.92 |
| Westerland | 33 | 27 | 38 | 35 | 31 | 24 | 38 | 33 | 0.89 | 0.87 | 0.90 | 0.95 | 0.51 | 0.70 | 0.94 |
| Zingst | 30 | 24 | 35 | 33 | 30 | 22 | 36 | 33 | 0.53 | 0.81 | 0.76 | 0.81 | 0.46 | 0.58 | 0.86 |
| Payerne | 28 | 25 | 38 | 38 | 36 | 24 | 41 | 41 | 0.92 | 0.89 | 0.90 | 0.91 | 0.93 | 0.93 | 0.95 |
| Eastern Europe | | | | | | | | | | | | | | | |
| Iskrba | 27 | 25 | 41 | 37 | 34 | 26 | 42 | 40 | 0.37 | 0.83 | 0.38 | 0.44 | 0.56 | 0.45 | 0.56 |
| Zavodnje | 36 | 25 | 39 | 37 | 34 | 26 | 41 | 40 | 0.87 | 0.89 | 0.87 | 0.88 | 0.92 | 0.88 | 0.94 |
| Kovk | 36 | 23 | 39 | 37 | 34 | 26 | 39 | 40 | 0.90 | 0.89 | 0.91 | 0.89 | 0.92 | 0.92 | 0.96 |
| Kosetice | 31 | 25 | 36 | 31 | 30 | 21 | 36 | 37 | 0.74 | 0.88 | 0.75 | 0.82 | 0.88 | 0.85 | 0.86 |
| K Puszta | 26 | 23 | 37 | 36 | 33 | 17 | 37 | 36 | 0.84 | 0.91 | 0.80 | 0.83 | 0.89 | 0.92 | 0.90 |

**Table 3.** Annual RERER values for Europe (total for all European sub-regions) and the European sub-regions shown in Figure 1 for the CO tracer, CO and $O_3$.

| Model | Europe | NW | SW | E | Gr + Tu |
|---|---|---|---|---|---|
| | | CO50 tracer | | | |
| EMEP_rv48 | 0.48 | 0.49 | 0.49 | 0.40 | 0.60 |
| IFS_v2 | 0.41 | 0.43 | 0.39 | 0.35 | 0.55 |
| | | CO | | | |
| EMEP_rv48 | 0.64 | 0.68 | 0.61 | 0.57 | 0.71 |
| IFS_v2 | 0.51 | 0.55 | 0.47 | 0.44 | 0.60 |
| CHASER_re1 | 0.52 | 0.53 | 0.53 | 0.45 | 0.64 |
| CHASER_t106 | 0.50 | 0.52 | 0.50 | 0.43 | 0.62 |
| OsloCTM3_v2 | 0.44 | 0.49 | 0.43 | 0.36 | 0.53 |
| CAMchem | 0.54 | 0.57 | 0.55 | 0.46 | 0.62 |
| GEOS-Chem | 0.41 | 0.43 | 0.24 | 0.35 | 0.56 |
| GFDL_AM3 | 0.51 | 0.54 | 0.49 | 0.53 | 0.60 |
| **model mean** | 0.51 | 0.54 | 0.48 | 0.45 | 0.61 |
| | | Ozone | | | |
| EMEP_rv48 | 0.87 | 1.01 | 0.80 | 0.81 | 0.76 |
| IFS_v2 | 1.12 | 1.38 | 1.04 | 1.10 | 0.83 |
| CHASER_re1 | 0.63 | 0.71 | 0.56 | 0.57 | 0.64 |
| CHASER_t106 | 0.64 | 0.74 | 0.56 | 0.58 | 0.63 |
| OsloCTM3_v2 | 0.89 | 1.06 | 0.80 | 0.91 | 0.71 |
| CAMchem | 1.02 | 1.38 | 0.87 | 1.09 | 0.71 |
| GEOS-Chem | 1.04 | 1.59 | 0.86 | 1.06 | 0.68 |
| GFDL_AM3 | 0.94 | 1.14 | 0.82 | 0.94 | 0.75 |
| **model mean** | 0.89 | 1.13 | 0.79 | 0.88 | 0.71 |

**Table 4.** Percentage contributions (where BASE - GLOALL represents 100%) to European annual ozone, summer (June,July, August) ozone, SOMO35 and $POD_1$ forest (SOMO35 and $POD_1$ forest only from the EMEP model) calculated from the 20% reductions of anthropogenic emissions in Europe, North America and East Asia. Model EMEP is EMEP_rv48, CAMC is CAMchem, GEOS is GEOS-Chem, IFS2 is IFS_v2, OSLO is OsloCTM3_v2 and CHAS is CHASER_re1.

| | EMEP | CAMC | GEOS | IFS2 | Oslo | CHAS |
|---|---|---|---|---|---|---|
| **EURALL** | | | | | | |
| Annual | 16 | 2 | -4 | -48 | 11 | 37 |
| Summer | 41 | 48 | 47 | 35 | 38 | 55 |
| SOMO35 | 31 | | | | | |
| PODy | 37 | | | | | |
| **NAMALL** | | | | | | |
| Annual | 20 | 19 | 23 | 24 | 21 | 11 |
| Summer | 13 | 8 | 24 | 27 | 13 | 6 |
| SOMO35 | 15 | | | | | |
| PODy | 14 | | | | | |
| **EASALL** | | | | | | |
| Annual | 26 | 15 | 18 | 22 | 14 | 9 |
| Summer | 15 | 7 | 16 | 27 | 11 | 4 |
| SOMO35 | 10 | | | | | |
| PODy | 17 | | | | | |

**Table 5.** Models to measurements bias in percent for 18 European CO sites and 113 European ozone sites. RERER: deviation from model average in percent. Percentage deviations more than 15% preceded by +/- signs in bold.

| | | Concentration | | | | RERER | |
| | | CO | | O$_3$ | | | |
| Model | CO Tr. | bias | Corr. | bias | corr. | CO | O$_3$ |
|---|---|---|---|---|---|---|---|
| EMEP_rv48 | yes | **-16** | 0.87 | **+ 19** | 0.75 | **+ 25** | -2 |
| IFS_v2 | yes | 1 | 0.82 | **-18** | 0.66 | 0 | **+26** |
| OsloCTM3_v2 | no | **-19** | 0.82 | **- 22** | 0.59 | -14 | 0 |
| CHASER_re1 | no | **- 24** | 0.80 | 10 | 0.66 | 2 | **- 29** |
| CAMchem | no | **- 25** | 0.80 | 22 | 0.73 | 6 | 15 |
| GEOS-Chem | no | **- 22** | 0.85 | 14 | 0.69 | **- 20** | 17 |
| GFDL_AM3 | partially | -13 | 0.77 | | | 0 | 6 |

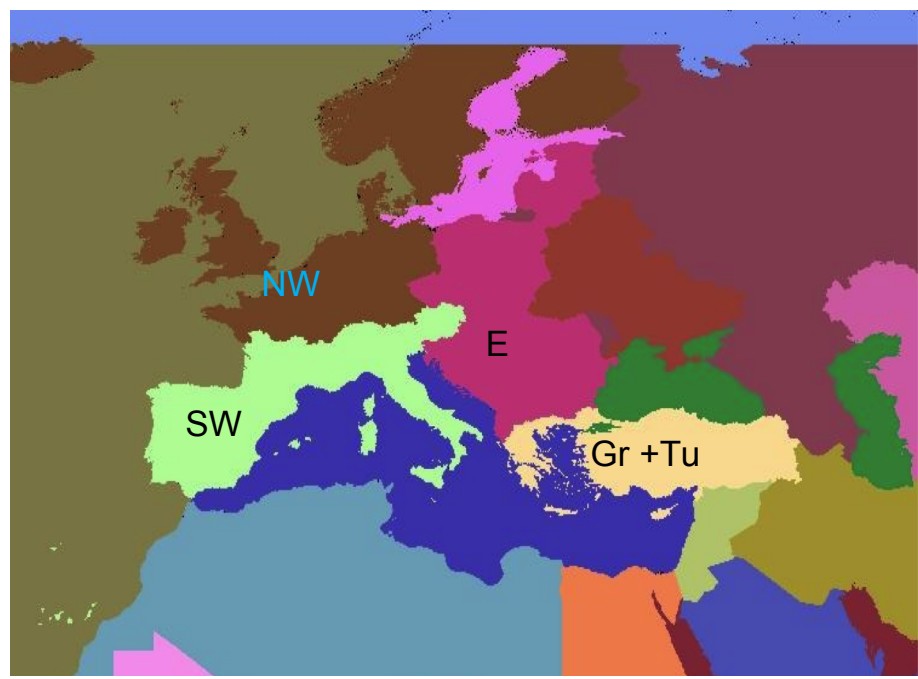

**Figure 1.** HTAP2 regions. The European land areas are further subdivided as: NW – Western Europe north of the Alps. SW – western Europe south of the Alps. E – eastern Europe. Gr + Tu – Greece and Turkey.

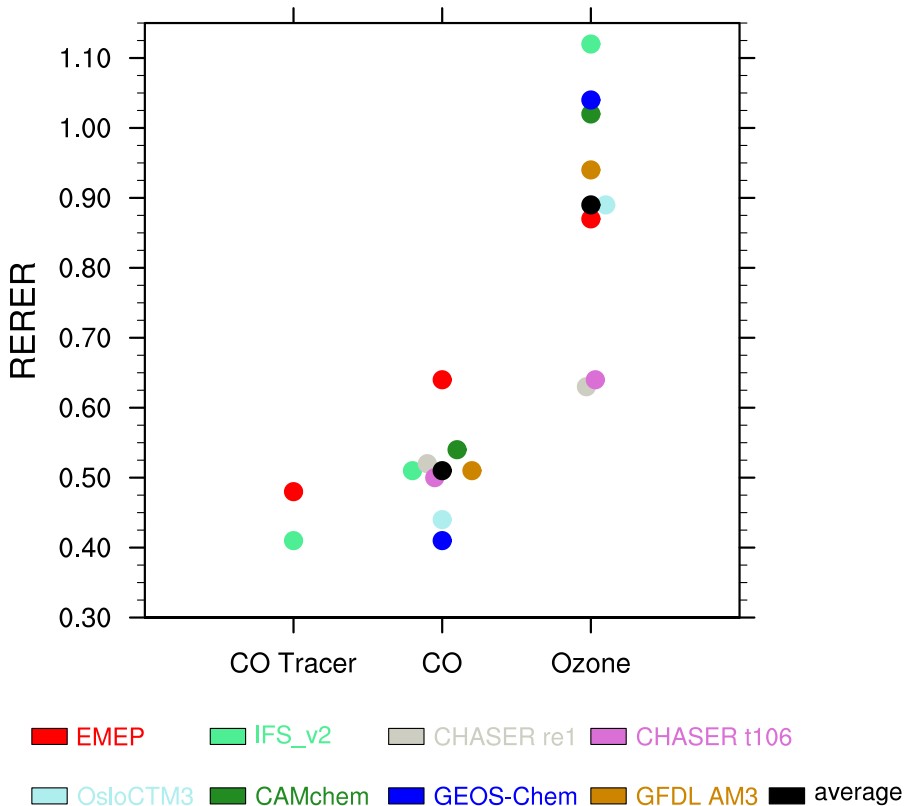

**Figure 2.** Model calculated annual CO tracer, CO and ozone RERER (Response to Extra-Regional Emission Reductions) values for Europe calculated by the models, see equation in section 4. Similar RERER values have been displaced horizontally.

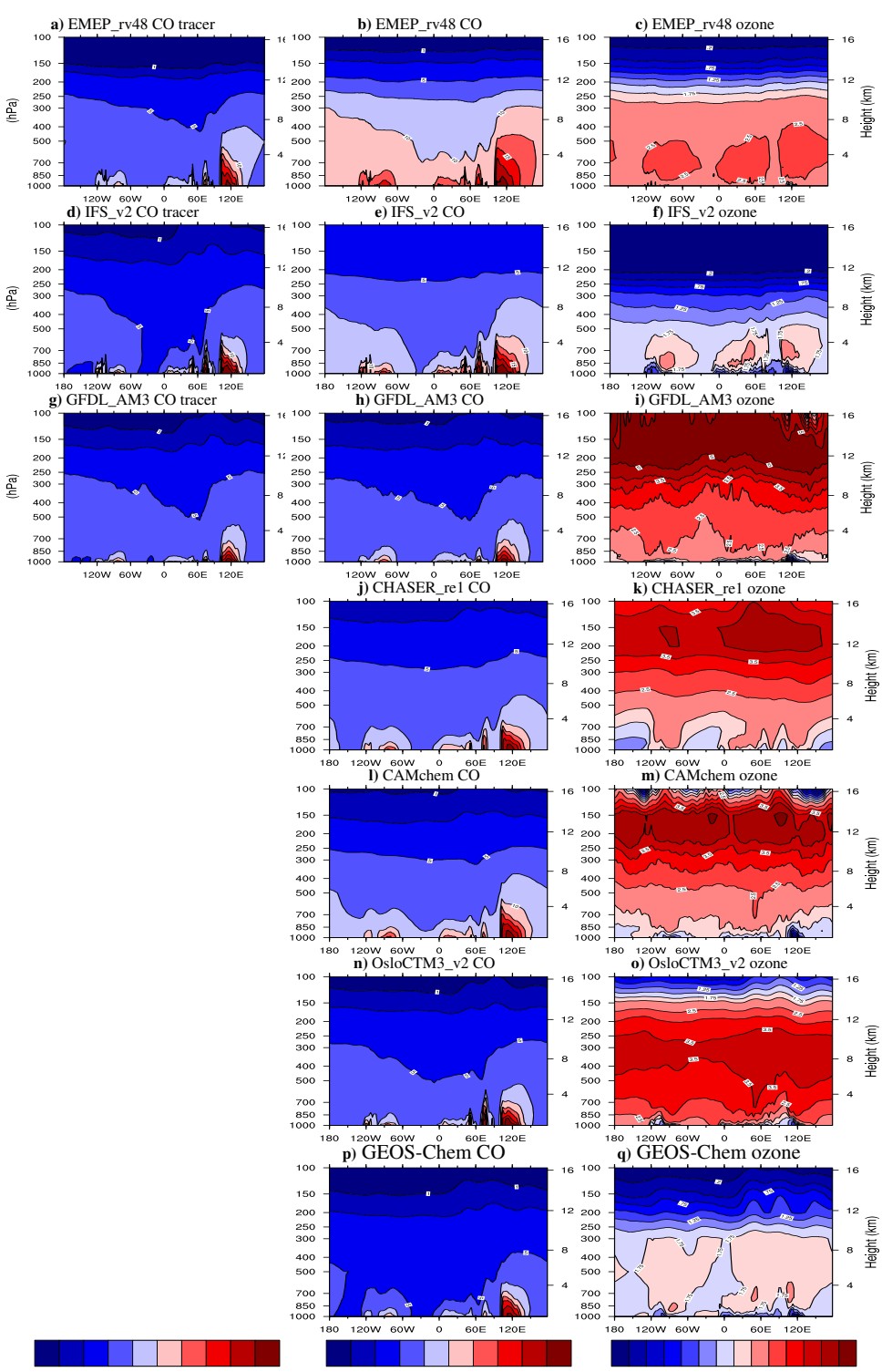

**Figure 3.** Annual contributions from the 20% (BASE – GLOALL) perturbations of the anthropogenic emissions to CO50 tracer (a,d,g), CO (b,e,h) and O$_3$ (c,e,f) in ppb zonally averaged between 30 and 60 deg. N. The models have been interpolated to a common vertical grid.

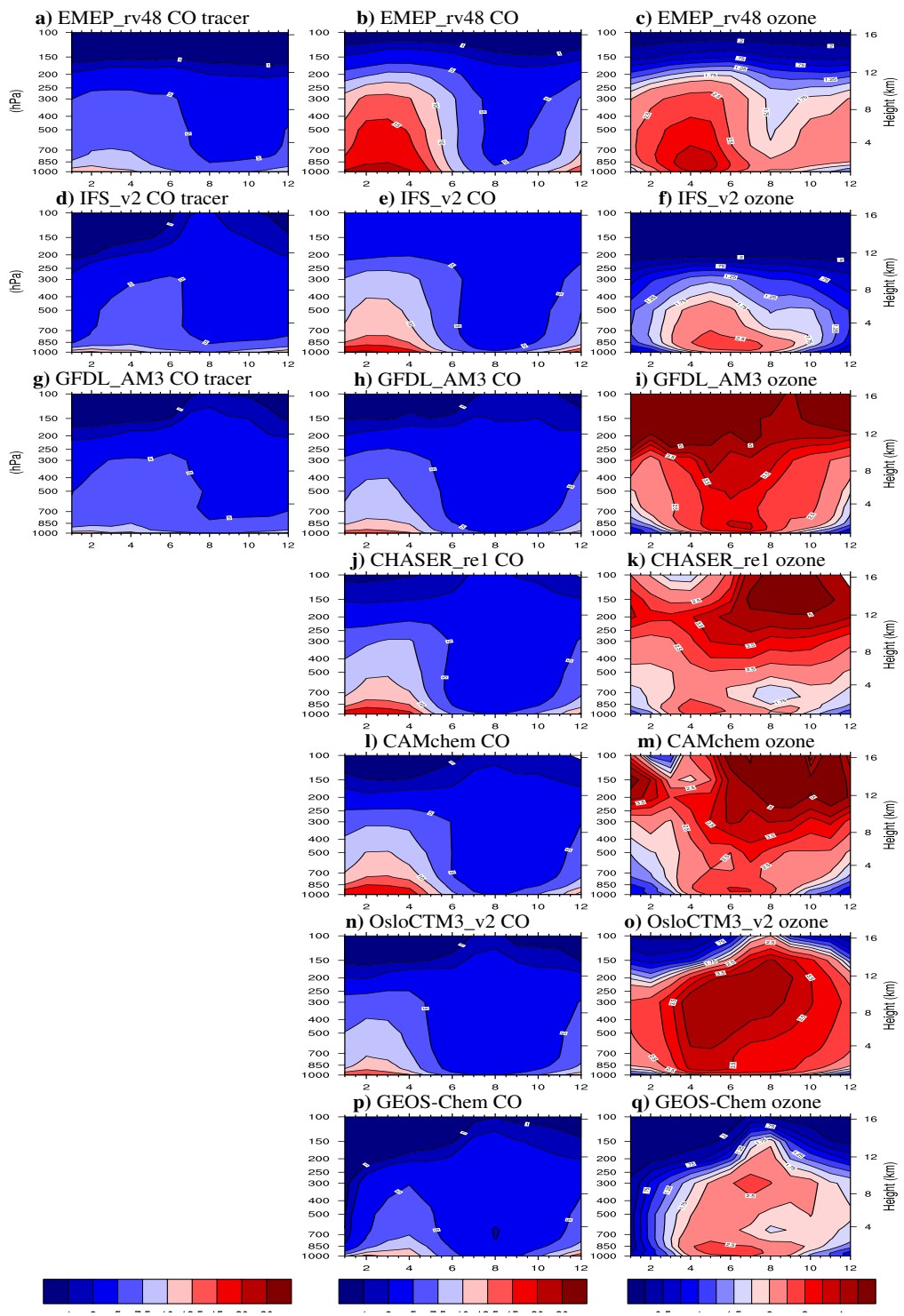

**Figure 4.** Monthly contributions from the 20% (BASE – GLOALL) perturbations of the anthropogenic emissions to co50 tracer (a,d,g), CO (b,e,h) and O$_3$ (c,e,f) in ppb averaged for the area bounded by 10°W to 35°E and 30 to 60 °N. The models have been interpolated to a common vertical grid.

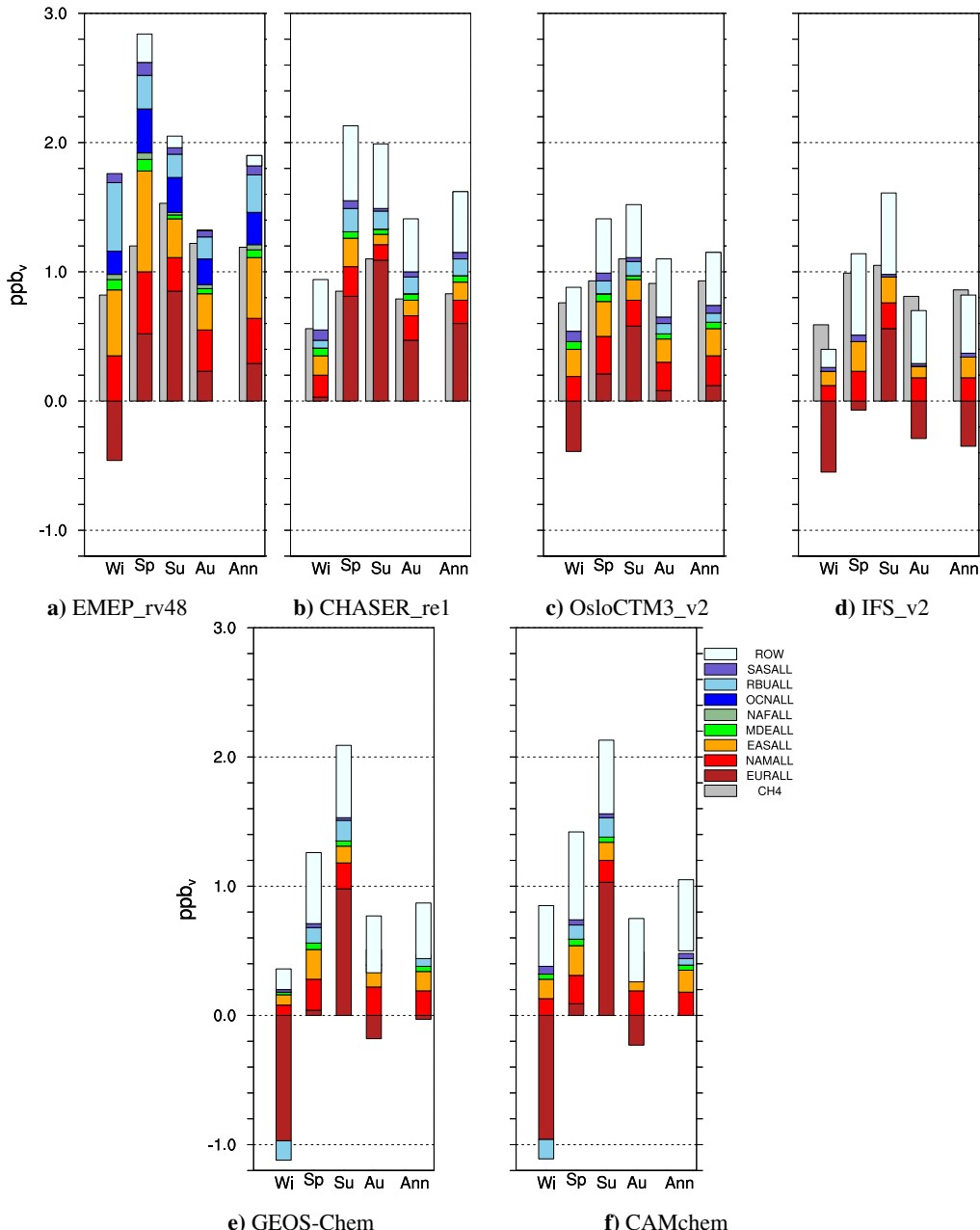

**Figure 5.** Contributions to European ozone levels (in ppb) from different world regions. (WI is December, January, February. SP is March, April, May. SU is June, July, August. AU is September, October, November). Note that the separate contribution from North Africa (NAFALL) and ocean shipping (OCNALL) is only included in the EMEP_rv48 model calculations. The Middle East (MDEALL) and Russia, Belarus and Ukraine (RBUALL) is not included in the IFS_v2 model. For all models contributions from missing regions are included as ROW (rest of the world). Note that the areas included in ROW is model dependent. For the four top row models the effects of a 20% increase in $CH_4$ is shown as a separate bar.

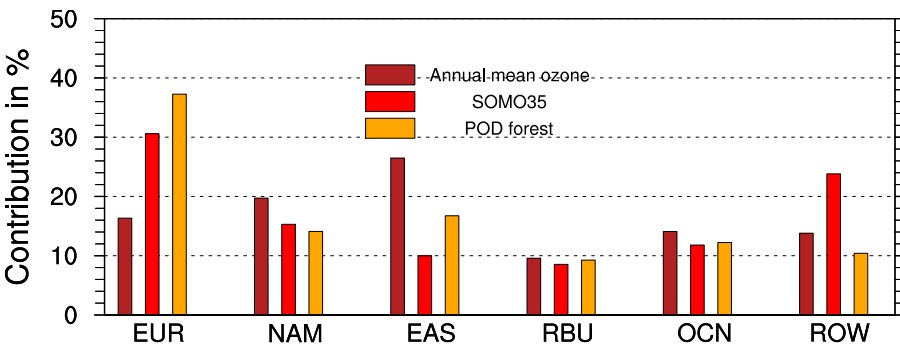

**Figure 6.** Contributions to ozone metrics annual mean ozone, SOMO35 and $POD_1$ forest in percent (where BASE - GLOALL represents 100%) as calculated by the EMEP_rv48 model. The metrics have been scaled so that the difference between the the BASE - GLOALL (20% anthropogenic emission reductions) calculations is 100% (the sum of EUR, NAM, EAS, RBU, OCN and ROW adds up to 100%).