# Peer review of "The effects of intercontinental emission sources on European air pollution levels"

_Atmospheric Chemistry and Physics, 2018_

## Referee Comment (RC1) · Anonymous Referee #2 · 6 Mar 2018

title: The effects of intercontinental emission sources on European air pollution levels

Overview: This paper has a significant contribution, but the manuscript has many errors and organizational issues. The authors approach of tracer, CO, ozone analysis could provided a nice insight into model differences. The manuscript, however, is full of obvious errors that should not make it to a reviewer.

The evaluation in a stepwise approach provides useful insight. Perhaps the most interesting results (rows 1-3 in Figure 3) however are not correctly displayed. This leads to difficulty interpreting *the* most interesting contribution.

There are many typos, grammatical errors, and glaring omissions. As a funny example, the HTAP phases are inconsistently numbered and the AQMEII project is misspelled.

[Figure]

The figures are incorrectly referenced and in some cases contain the wrong content (e.g, fig 3i). Several of the models are not referenced when they are clearly relevant to a point the author is making. For example, where is AM3 on page 8 lines 240-245?

The manuscript needs major reorganization. The manuscript reads as though sections were added without consideration of what had been previously said. For example, the authors suggest that evaluation should be found in other papers which leads the reader to believe evaluation will not be discussed (section 3: Models vs measurements) and then simplistic evaluation is provided in the discussion (section 5). The organization provides little methodological context for the results in Section 2 (the HTAP2 model setup) – instead, "revealing" methodological inconsistencies in the results (e.g., Section 4.3) or model configurations in the discussion (Section 5). Section 5 is a combination of nice insights and throw away paragraphs that seem unfinished.

Lastly, the issue of titration seems important to the conclusions. The results in Figure 5 show that most models calculate local titration in many months. While that suggests low sensitivity, "contribution" as defined by source apportionment or tagging would likely produce a much larger local contribution estimate. The 20% maybe just beyond removing titration but before reaching control effectiveness. In the conclusions, the authors suggest that controls have been offset by increases in other regions. The titration still present, however, suggests that European controls have also been offset by removed local suppression. These are important considerations for highly non-linear regions like Europe that are not sufficiently addressed.

This manuscript is not ready for publication. The underlying work is clearly important and makes a contribution, but the presentation (including the writing and organization) are not ready for publication.

Specific Comments (lines: comment)

16: capitalization error. 25: missing parenthesis 29: verb agreement 46-54: The list of published papers should be used to provide context. Here it is simply a list. 60: details

like model count would be better in the methods. 68-70: differ should be differences? 72-74: poorly written. 82: "etc" seems particularly poor when later you will refer to advection schemes as a causal difference. 87: missing space 92: missing space 95: missing "is" 95: How does evaluation of upwind sources affect conclusions about transport to Europe? 99: spell out GAW 101: How "high" correlations are expected given the resolutions of the models? 101-102: resolutions of all the models should be provided in the methods rather than the comparison to measurements. 98-109: How is it that CO deserves a site-by-site comparison and ozone 112: The authors should mention that they do have some surface evalaution in this paper. Currently, Table 3 in this manuscript is not referenced until Section 5. 114: There is currently no discussion of ozone results except to say they exist somewhere in the supplement. Why is this sufficient? 122-123: There must be more discussion of the basic results that will clearly affect transport. 138: Here and elsewhere the definition of regions is incorrect. Here you have NW, SW, SE, GR+TU. In the Figure, you have NW SW, E, GR+TU. Other places you have NW, SW, E, SE. Choose one, and be consistent. 139-140: Is this source apportionment the same as contribution in sections 4.4 and 4.5? 142: rate of decay is later explained, but here seems completely arbitrary. 182: Wrong figure references. 185-189: The reasonableness of this should be discussed. 205: This gets discussed in several places and is really part of the methods. 217-219: Web citation is inappropriate. Further, the lifetime of ozone is expected to vary with respect to season and altitude (Wang et al. 1998; Brasseur, Orlando, and Tyndall 1999). Estimates of lifetime at 500hPa range from 15-160d and from 40-300d at 10km. Your upper bound of 18days is misleading. Table 1.1 of the HTAP 2010 report cites weeks to months in the free troposphere. The IPCC range of values do not acknowledge the complexity of ozone transport. 242: AM3? 246-247: Provide some reference or evidence. 247: here = PBL? 254-284: Is this contribution from a simple mean within seasons? What months were included in each season? Are the numbers in the text ensemble means? What about ensemble mean RBU? MDE? EU? 290-291: Did they "too" calculate smaller "than in this study" or did they "too" calculate "smaller as in this study"? 269: MDE

appears to always be small. 272-273: Did these other studies use the same model? 277: missing parenthesis 274-280: methods? 305: HTAP1? 306-335: There needs to be a clearer connection to the previous section. In fact, you could just add two bars to Figure 5a. That would help to connect the of POD and SOMO35 to the seasonality of titration. 361-371: Terse and uninformative. 390-392: See previous comments about ozone lifetime. 400: probably? 405: Ozone?

Table 1: If mountain sites are used at readers peril, consider making room for ozone evaluation by moving them from the first data result.

Table 2: Update region definitions to be consistent with figures and text.

Figure 1: update region names to be consistent. Also, too many extra colors so it is hard to tell what is included. Is the Baltic Sea part of Eastern Europe? Black Sea? Caspian Sea? Mediterranean?

Figure 2: necessary?

Figure 3: lettering needs to be updated in the figure and in the text. What was the common grid and how was it treated when a gridcell at 1000hPa was below the surface?

Figure 3: 3i is AM3 CO not ozone. Column 3: consider a scale that does not saturate in so much of the figure.

Figure 4: North and south boundaries are unnecessarily different from figure 3. Further, this highlights that no meaningful discussion of the boundaries was made. In fact, 50E includes a lot of Russia and a lot of ocean. Column 3: consider a scale that does not saturate in so much of the figure.

Figure 5: There is no discussion about the CHASER model being the only one without apparent titration, and this should be discussed somewhere. Region definitions should be consistent with the text or the text should be consistent with the figure. The units are cutoff on the first row.

Figure 6: region definition nomenclature. I recommend showing as 3 stacked-bars (or adding to Figure 5). If I am interpreting this right, the RAIR is 84% compared to 43% from HTAP1. I suspect that all models provided annual and I think reporting RAIR would be useful (maybe in Figure 2).

References: Brasseur, Orlando, and Tyndall 1999, ISBN:978-0-19-510521-6 Wang et al., 1998 doi:10.1029/98JD00156 Figure 4 HTAP, 2010

---

## Referee Comment (RC2) · Anonymous Referee #1 · 19 Apr 2018

Review of The effects of intercontinental emission sources on European air pollution levels, submitted to Atmos. Chem. Phys by Jonson et al., 2018

The manuscript submitted by Jonson and co-authors address the issue of the relative importance of local (continental in this case) and remote (hemispheric) sources to surface ozone levels. It builds upon the modelling works undertaken by the UNECE Task Force Hemispheric Transport on Air Pollutants. The questions addressed by HTAP have a clear political relevance. It also relies on a vast amount of scientific work well illustrated by the number of articles in the current special issue. In fact, the topic chosen by Jonson et al. is perhaps one of the most relevant, at least from a European perspective. Unfortunately, the article as it stands today is not of satisfactory quality to allow publication in ACP. Major revisions are needed, and my review below explains

[Figure]

why. The approach overall scientifically sound, but there are a number of omissions that must be addressed. Also, the quality of the presentation is far below the usual ACP standards. Nevertheless, for the reasons listed above in terms of motivation for the study, the authors should be offered a chance to resubmit their work in a revised form.

Specific comments

The aim of the article and its structure are not well explained in the introduction. The structure in L66-70 seems not to match the actual content of the section (Section 5 became a very superficial discussion on model resolution, it seems that the authors forgot that they originally expected to suggest improvement in the experiment design in that section). It is unclear why comparison with measurements come back in Section 5, while it was introduced in Section 3. At the end of the introduction the reader is already sceptical to what extent the paper will address the problem at hand.

L23: define here the CTM acronym, which usually refers to chemistry transport models rather than chemical tracer models

L25: TF-HTAP is organized under the EMEP programme of LRTAP

L50: add that the region targeted in that paper is Europe, but that (unlike in HTAP) the contribution is assessed by model tagging rather than sensitivity experiments

L53-54: the sentence on additional papers is not relevant, suggest removing

L60-63: should be moved to the experiment description part (ex: L78)

L63: rephrase "secondly we look at CO" to better introduce the actual chemical compound in opposition to the CO-like tracer

L87: For transparency and reproducibility concerns, but also with regards to the HTAP requirements, GFDM_AM3 should not be included if it is not part of the database.

L92: The Galmarini et al. article in the special issue is focused on the complementarity

of global and regional models rather than model evaluation. In the version currently in discussion, only a Taylor diagram is given with models not labelled. Therefore it cannot be considered as a satisfactory reference regarding the capability of HTAP models in capturing surface ozone. Such an analysis should be included here if not covered elsewhere. The scatter plots in supplementary material is a good start, but further discussion is needed. The selection of Airbase sites is very questionable at this scale.

L104: a reference is needed to conclude that GAW sites are affected by local sources. Similarly, one could question why engaging in such analysis if global models are not capable to capture regional sources. If the author conclude that it is the case, it would be a major conclusion of the paper.

L116: what is the source of ozone profiles ?

L118: how "approximate" is the temporal matching between model and observations?

L123: more quantitative results are needed to support the "tendency" for underestimation in tropospheric summertime ozone.

L154: why is GFDL_AM3 included in Figure 3 for CO_tracer but not in Figure 2?

L159: why would EMEP have a too strong convection ? If the comparison with measurement suggest that EMEP performs better than other models (L348), maybe the other models have a too weak convection ?

L162: the larger vertical mixing seems to occur mainly in winter for EMEP, isn't that conflicting with the hypothesis about the role of convection ? Maybe more discussion would be needed on the vertical diffusivity and resolution of the various models.

L183: The difference between CO_tracer and CO seems larger for EMEP than for the other models. Would it also be the case in terms of relative increase, and if so, why?

L195: the model differences for OH are very impressive (Fig 9 of the supplementary material). To the extent that one may wonder the relevance (and need) to produce
a multi-model mean. Further discussion and external references are needed for that section. The sensitivity to upper boundary conditions, especially for EMEP that seems to behave differently.

L213 : a reference is needed to support the statement on the relative contribution of stratospheric/tropospheric ozone.

L245: the discussion on aircraft emissions is interesting, but it seems that there are more important differences, such as the role of surface titration (why EMEP seems the less sensitive despite the higher resolution). Or the fact that the O3 response of Chaser is actually very close to that of CO_tracer (or is it a mistake in the Figure?)

L256: the fact that CH4 is excluded from the experiments should appear before in the experiment description (Section 2).

L259: the explanation of figure 5 needs to be re-worked in the text, in particular to explain that even if some sources are not isolated in some models, their contribution is still accounted for in the "remaining" fraction.

L266: more quantification is needed regarding the relative role of external/European sources. Figure 5 indicates that the external contribution seems indeed to exceed European contribution, but they are actually not that far. The percentage contribution (with error bar) should be given.

L275: the comparison the HTAP1 is too weak. It is very frustrating not to better understand the added value of the new experiment and to what extent the earlier conclusions still hold. The benefit of having engaged in a complete new experiment should be better assessed. For instance by looking at a subset of models having participate to both and investigating clusters of regions for the reference/sensitivity simulations to conclude on the importance of (i) emission changes, (ii) region definition, (iii) participating models.

L301: it is surprising to say that a comparison should not be made, when it is made by the authors. There were no similar discussions on the realistic aspect of sensitivity ex-

periments for non-CH4 species, so from an academic perspective regarding chemical sensitivity the comparison does hold.

L330: the results related to ozone indicators are interesting and worth being highlighted in the abstract. It is frustrating that only one model can be used here, especially given the apparent different behaviour with regards to titration. More efforts should be given to investigate the HTAP database in order to include more models for a comparison of summertime mean of daily ozone maxima, or at least summertime mean ozone.

L350: according to Section 3.2, EMEP is not the only model to display an overestimation of tropospheric ozone.

L360: how can a lower diffusion can lead an underestimation of surface CO, the opposite would be expected

L367: what is the rationale for a relaxation to zero in the GEOS-Chem adjoint?

L372: Section 5 is very descriptive and lacks a clear outcome

L399: From the results provided in Section 5, it can not be concluded that the results are not sensitive to model resolution.

L406: in Section 4.1 it is rather convection that is put forward rather than chemistry. The following sentence (L407) also goes in that direction. It is quite surprising to read that the conclusion and the content of the paper seem contradictory.

L419: from Figure 5, it seems that about half of ozone can be mitigated with European sources, isn't that "sizeable" already? More precise quantifications must be given in the conclusion and abstract with key figures and associated error bar across the multimodel ensemble.

Table 1: what is given on the right part of the table? From the text, it appears to be correlation, but that should be stated clearly. Is CHASER_rel actually Chaser_t42 according to table 1 of the supp. Mat. A uniform model labelling would be appreciated.

Table 2: ibid about CHASER_rel. The labels of regions should be consistent with Fig 1.

Table 3: swap the first and second sentences. Use boldface rather than larger signs for important deviations.

Figure 2: GFDM_AM3 missing for CO_Tracer

Figure 3: panels g) and i) appear identical

Figure 4: the panels are truncated

Figure 5: the panels are truncated

Figure 6: add that the results to 20% perturbation are plotted.

Supp. Mat. Table 1: all models are in bold, not the first 7, since 7 models are displayed. The information about spin up should be given in the experiment description, not in Table 1. Footnote #2 is not references in the text.

Supp. Mat. Figure 1: What happened in Heimaey in May in observations?

Supp. Mat. Figure 4: More details are needed in the legend about the source of data and the indicator displayed

Editing

L26: dot missing after the parenthesis

L42: define GCM, isn't it rather Climate Chemistry Models that would be referred to ?

L48: organic aerosols are OA

L50: problem with the reference

L73 dot missing after the parenthesis

L73, L75: dot missing after etc.

L87 : space missing before GFDL

L92: space missing after measurement

L111: AQMEII

Hyperlinks should be in footnotes rather than in the text

L360: lower cap for underestimates

L366: the notation "20+ percent" is not appropriate for a scientific paper

L369: incomplete sentence

L402: d missing in "and also"

L409: "in" missing: "steps in improving"

L435: EMEP and Airbase data should also be acknowledged here.
* * *

---

## Author Comment (AC1) · 8 Jun 2018

We thank the reviewer for the effort to see into this multi-author paper. We apologise for oversights, partly due to the complex nature of the multi-model evaluation.

General comments

Lines 66-70: The structure in L66-70 seems not to match the actual content of the section (Section 5 became a very superficial discussion on model resolution, it seems that the authors forgot that they originally expected to suggest improvement in the experiment design in that section). It is unclear why comparison with measurements come back in Section 5, while it was introduced in Section 3. At the end of the introduction the reader is already sceptical to what extent the paper will address the problem at

hand.

We have changed this part of the paper, including a better motivation for Section 5 (in addition section 5 has also been improved). The motivation for section 5 is to sum up the results for the individual models and as such make the reader better prepared for the conclusions. These lines now reads: In this paper we first briefly discuss the model comparison to measurements in section 3. In section 4 we go on to describe the source receptor relationships for Europe, including a discussion on how and why the model results differ. Finally, in section 4 we sum up the results for the individual models. Based on model performance compared to measurements and where and when deviations in model results compared to the other models occur we try to indicate the origins of the differences in model behaviour. In the conclusions we then suggest some directions on how this information could be used to harmonize and improve future model calculations.

Specific comments

Line 23: define here the CTM acronym, which usually refers to chemistry transport models rather than chemical tracer models

Added CTM acronym

Line 25: TF-HTAP is organized under the EMEP programme of LRTAP

Added that HTAP2 reports to the EMEP stearing body

Line 50: add that the region targeted in that paper is Europe, but that (unlike in HTAP) the contribution is assessed by model tagging rather than sensitivity experiments.

We have added that tagging is used in his model.

Line 53-54: the sentence on additional papers is not relevant, suggest removing

We have removed the list and replaced in with a reference to the acp special issue.

Line 60-63 should be moved to the experiment description part (ex: L78)

This sentance is removed. This discussion is included elsewhere.

Line 63: rephrase "secondly we look at CO" to better introduce the actual chemical compound in opposition to the CO-like tracer.

We have changed to: Secondly we investigate CO as an interactive component of the atmosphere, participating in chemical reactions.

Line 87: For transparency and reproducibility concerns, but also with regards to the HTAP requirements, GFDM_AM3 should not be included if it is not part of the database.

We will now included GFDM_AM3 results in the database, note that these model results are present in a slightly different format. We have chosen to include these results as so few models have uploaded relevant results for this study.

Line 92: The Galmarini et al. article in the special issue is focused on the complementarity paper of global and regional models rather than model evaluation. In the version currently in discussion, only a Taylor diagram is given with models not labelled. Therefore it cannot be considered as a satisfactory reference regarding the capability of HTAP models in capturing surface ozone. Such an analysis should be included here if not covered elsewhere. The scatter plots in supplementary material is a good start, but further discussion is needed. The selection of Airbase sites is very questionable at this scale.

See also comments to reviewer 2 for this point. We have included ozone time series for several GAW sites in the supplementary material. Furthermore we have included a table with statistics similar to what is already included for CO. The mentioning of AirBASE as data source was wrong. The scatterplots in the supplementary material are based on European rural and remote sites from EMEP, as stored in the EBAS database.

Line 104: a reference is needed to conclude that GAW sites are affected by local

sources. Similarly, one could question why engaging in such analysis if global models are not capable to capture regional sources. If the author conclude that it is the case, it would be a major conclusion of the paper.

We agree with the reviewer that GAW sites are selected to represent regional background conditions. The argument was removed.

Line 116: what is the source of ozone profiles?

The original source for the ozone sondes is the "World Ozone and Ultraviolet Radiation Data Centre". This information is now included in the text. Data providers have been contacted and offered co-authorship.

Line 118: how "approximate" is the temporal matching between model and observations?

In HTAP2 model profiles are provided on an hourly basis. We have added "(to the nearest hour)" in the text.

Line 123: more quantitative results are needed to support the "tendency" for underestimation in tropospheric summertime ozone.

We have extended the section with the interpretation of the ozone profiles. This part now reads:

With a relatively inactive chemistry in the winter months the measured ozone profiles show little vertical variability, with ozone mixing ratios in the troposphere increasing gradually with height. Model calculated ozone profiles are close to the measurements. As the chemical activity increases in Spring and summer months the vertical variability increases, reflecting air masses of significantly different photochemical history at different levels. As was shown in Jonson (2010) the models are not capable of reproducing this vertical structure in ozone levels. Most of the models underestimate free tropospheric ozone in the summer months.

[Figure]

Line 154; why is GFDL_AM3 included in Figure 3 for CO_tracer but not in Figure 2?

Figure 2 requires data from the BASE, GlOALL and EURALL scenarios, whereas Figure 3 is based on the BASE and GLOALL scenarios only. For the GFDL_AM3 model we have no data for the CO tracer for the EURALL scenario. A comment has been added in the text.

Line 159: Why would EMEP have a too strong convection? If the comparison with measurement suggest that EMEP performs better than other models (L348), maybe the other models have a too weak convection ?

We have deleted the word too, subsequently changing the meaning so that we now say that the EMEP model has strong convention and not "too" strong convection.

Line 162: the larger vertical mixing seems to occur mainly in winter for EMEP, isn't that conflicting with the hypothesis about the role of convection? Maybe more discussion would be needed on the vertical diffusivity and resolution of the various models.

Running the global EMEP model with and without spinnup we see marked differences (in ozone) lasting all the way into Spring. We believe that this is caused by ozone lifted into the free troposphere increasing the free tropospheric reservoir of ozone in the following winter and spring. We believe this also the case for the CO tracer.

The text has been made more clear on this point: Differences in mixing ratios peak in the first part of the year when emissions are high and the exchange between the boundary layer and the free troposphere over Europe is weak. Differences in the free troposphere may reflect CO tracer advected from regions upwind with convective activity also in winter, or in the preceding autumn months increasing the free tropospheric reservoir in the following winter and spring.

Line 183: The difference between CO_tracer and CO seems larger for EMEP than for the other models. Would it also be the case in terms of relative increase, and if so, why?

[Figure]

The widening gap between EMEP end the IFS model is commented below in the same section. It is attributed to the possible lower OH levels in the EMEP model compared to the IFS model. (even though OH is not provided by the IFS model).

Line 193: the model differences for OH are very impressive (Fig 9 of the supplementary material). To the extent that one may wonder the relevance (and need) to produce a multi-model mean. Further discussion and external references are needed for that section. The sensitivity to upper boundary conditions, especially for EMEP that seems to behave differently.

A more detailed discussion on differences and of the effects of OH is now included in several places in the manuscript.

Line 213: a reference is needed to support the statement on the relative contribution of stratospheric/tropospheric ozone.

References to the HTAP 2010 report and Stevenson et al. 2006 now included.

Line 245: the discussion on aircraft emissions is interesting, but it seems that there are more important differences, such as the role of surface titration (why EMEP seems the less sensitive despite the higher resolution). Or the fact that the O3 response of Chaser is actually very close to that of CO_tracer (or is it a mistake in the Figure?)

Discussion on aircraft emissions strengthened following comments also from reviewer 2. Regarding the GFDL_AM3 (not CHASER), the figure is corrected. We believe that the reason why the EMEP model is less sensitive to titration must be sought in the chemistry schemes. This is now discussed later in the paper (see also comments from reviewer 2).

Line 256: the fact that CH4 is excluded from the experiments should appear before in the experiment description (Section 2).

The statement that CH4 is not included in the experiment is moved to section 2.

[Figure]

Line 259: The explanation of figure 5 needs to be re-worked in the text, in particular to explain that even if some sources are not isolated in some models, their contribution is still accounted for in the "remaining" fraction.

This section now reads: For each model the contribution from ROW (Rest Of the World) is calculated by subtracting the sum of the contributions from from available world regions from the BASE - GLOALL contribution. Thus the portion related to ROW includes a varying mixture of world region definitions depending on the model.

Line 266: more quantification is needed regarding the relative role of external/European sources. Figure 5 indicates that the external contribution seems indeed to exceed European contribution, but they are actually not that far. The percentage contribution (with error bar) should be given.

We have added an additional table listing the percentage contributions to annual ozone and summer ozone to Europe from Europe, North America and East Asia. In addition we also list the contributions to SOMO35 and POD forest calculated by the EMEP model. The numbers are a subset of those displayed in Figure 5.

Line 275: the comparison the HTAP1 is too weak. It is very frustrating not to better understand the added value of the new experiment and to what extent the earlier conclusions still hold. The benefit of having engaged in a complete new experiment should be better assessed. For instance by looking at a subset of models having participate to both and investigating clusters of regions for the reference/sensitivity simulations to conclude on the importance of (i) emission changes, (ii) region definition, (iii) participating models. Conclude on emission changes:

The paragraph has been rewritten and now reads: In comparison to HTAP1, HTAP2 regions are better defined. In addition emissions as well as models are up-to-date. To disentangle whether the changes from HTAP1 to HTAP2 are due to emissions, a changed model ensemble or changes in receptor regions is unfortunately not possible in a fully quantitative way. Source and receptor regions have been chosen in HTAP2

to cover the land-only politically connected regions accurately on a 0.1 degree grid. In HTAP1 the EUR region was a simple latitude - longitude box, also including parts of North Africa, the Middle East, Russia, Belarus, Ukraine and large sea areas, all of these identified as non European regions in HTAP2. In HTAP2 the European region is smaller, thus exporting larger fractions to nearby regions, but most major HTAP1 source regions are located within the smaller HTAP2 region, thus making this region more sensitive to titration effects. As a result the effects of emissions on ozone levels from the EUR region to itself is reduced.

The ensemble mean contribution to ozone levels from Europe to itself has decreased from 0.82 +- 0.29 ppb in HTAP1 to just 0.11 +- 0.32 ppb in HTAP2. Also - total and regional distribution of emissions for the base year changed from HTAP1 (2001) to HTAP2 (2010). Gaudel al. (2018) have analysed the ozone trends between the years 2000 and 2014. Over Europe. They found a general ozone increase in the winter months (December, January, February) and a general decrease in the summer months (June, July, August). The emission trends in the HTAP1 world regions are given in Turnock et al. (2018) between 2001 (the base year for HTAP1) and 2010. The changes in measured ozone are consistent with the reductions in European (and North American) emissions of NOx (along with other ozone precursors) over the same period resulting in less titration and thus increased ozone levels in some areas mainly in the winter months, and simultaneously less net ozone production in summer. Likewise emissions in North America have decreased and may explain the 0.37 +- 0.10 to 0.22 +- 0.07 ppb decrease in the ensemble mean contributions from North America to European ozone levels. Over the same period emissions in other world regions as East Asia have increased. This increase may explain the 0.17 pm 0.05 to 0.22 +- 0.13 ppb ensemble mean increase from HTAP1 to HTAP2 in the East Asian contribution to European ozone levels. Contributions from South Asia are small in both HTAP1 and HTAP2 (0.07 versus 0.05).

Line 301: it is surprising to say that a comparison should not be made, when it is

made by the authors. There were no similar discussions on the realistic aspect of sensitivity experiments for non-CH4 species, so from an academic perspective regarding chemical sensitivity the comparison does hold.

Thanks. The point is that the effects of direct changes in concentrations is not the same thing as changing the emissions. This is now clarified and rewritten: However, comparing a 20% change in $CH_4$ concentrations, and the effects of the GLOALL emission scenario requires careful interpretation. Because of its relatively long lifetime of the order of 10 years in the atmosphere, a 20% change in concentration corresponds to an approximate 40 year long historic CH4 trend (Meinshausen 2011). The GLOALL scenario is not accounting for the full impact of a continued 20% reduction in emissions. With a continued emission reduction scenario, the overall ozone reductions would be larger, while the methane attributable fraction, relatively, would be smaller.

Line 330: the results related to ozone indicators are interesting and worth being highlighted in the abstract. It is frustrating that only one model can be used here, especially given the apparent different behaviour with regards to titration. More efforts should be given to investigate the HTAP database in order to include more models for a comparison of summertime mean of daily ozone maxima, or at least summertime mean ozone.

Unfortunately hourly data with attribution to source regions are only available for the EMEP and the CHASER model. We have thus instead included a table comparing annual average and summertime ozone based on the numbers in Figure 5. (see comments to line 266). SOMO35 and POD forest from the EMEP model are included in the same table. As SOMO35 and POD are mainly added up in the summer months the percentages are similar in the EMEP model and the table gives an indication on the differences between the models.

In addition to the table we have added some additional text: The regional contributions, expressed by these metrics, are also listed in table (new table). The figure and table

clearly shows that the choice of metric matters, in particular for the effects of European Emissions. POD forest is accumulated in the growing season in summer. A large portion of SOMO35 is also accumulated in the summer months. Table (new table) also lists the percentage contributions to summer ozone for all models. The similarities in the percentages for summer ozone and the ozone metrics in EMEP\_rv48 is an indication that also for the other models these percentages are comparable.

Line 350: according to Section 3.2, EMEP is not the only model to display an overestimation of tropospheric ozone.

Yes, text is changed to: The model is one of the models with highest overestimation of ozone in the free troposphere in the winter and spring months.

Line 360: how can a lower diffusion can lead an underestimation of surface CO, the opposite would be expected.

Thanks - we agree. If the low RERER is caused by too much CO remaining in the PBL it should result in an overestimation of surface CO compared to measurements. We have deleted this statement.

Line 367: what is the rationale for a relaxation to zero in the GEOS-Chem adjoint?

The GEOS-Chem model has only ozone chemistry in the troposhere, and stratospheric levels should be disregarded.

Line 372: Section 5 is very descriptive and lacks a clear outcome

We have added more text, including text with additional motivation for this section: Here we try to point out if, and at what stage, the results from the individual models deviate from the other models. It should be stressed that such a deviation does not necessarily imply that the results from a particular model is wrong.

For the individual models we have tried to identify one ore more feature where the individual models differs from the other models.

Line 399: From the results provided in Section 5, it can not be concluded that the results are not sensitive to model resolution.

Yes, this is probably phrased too general - This part now reads: The model resolution differs between the individual models. Model results from the two CHASER models, differing in model resolution only, are qualitatively similar when compared to measured CO and O3} at background measurement sites and very similar in RERER for CO and ozone, suggesting that resolution differences at the scales investigated here, are not important to explain RERER differences between the global models. Still, it is difficult to conclude in general to what extent horizontal resolution affects the source receptor calculations at intercontinental scales.

Line 406: it is rather convection that is put forward rather than chemistry. The following sentence (L407) also goes in that direction. It is quite surprising to read that the conclusion and the content of the paper seem contradictory.

This part now reads: The joint and consistent analysis of a CO tracer, CO and O3 in this paper is a tool in understanding where and why (right or wrong) the models differ, however, it has a potential for wider use, enhancing our understanding of the result and also as a tool for model improvements, reducing the overall uncertainty in future model calculations. We believe that in order to close the gap in model results, and subsequently improving the reliability of the model output, possible future model inter-comparisons should be more process oriented (transport, depositions, chemistry etc). Our study shows that models differ already for CO and the inert CO tracer, where differences were established with 2 models, but that differences are amplified as more chemistry is added. Note that the CO RERER and O3 RERER values are not correlated taken the models as samples. The big additional spread in model results for ozone is clearly induced by differences in model chemistry and for instance treatment of titration in the winter boundary layer. However, differences in chemistry may well also be induced by differences in advection/convection as the level of exchange will inevitably affect the chemical regime in both the free trposphere and in the boundary

layer. We believe therefore that further process oriented evaluations (comparing advection/convection, chemistry, dry and wet deposition etc separately) should be made, making use of relevant meteorological and chemical measurements.

Line 419: from Figure 5, it seems that about half of ozone can be mitigated with European sources, isn't that "sizeable" already? More precise quantifications must be given in the conclusion and abstract with key figures and associated error bar across the multi-model ensemble.

We now include a comparison and discussion of RAIR in HTAP1 and HTAP2 in section 4.4. Furthermore RAIR is also included in the conclusions and the abstract.

The (almost) first parts of the conclusion section now reads: The model calculated relative contributions to surface ozone levels from non European sources is much larger in HTAP2 compared to HTAP1. Mainly because the contributions from Europe to it selves has decreased from 0.82 ppb to just 0.11 ppb. As a result RAIR has increased from 43 to 82\%. In parts differences could be explained by decreasing emissions in Europe and increased emissions in most other regions as East Asia from year 2001 to 2010. However, the results from the two HTAP phases can not easily be compared, .......

Table 1: what is given on the right part of the table? From the text, it appears to be correlation, but that should be stated clearly. Is CHASER_rel actually Chaser_t42 according to table 1 of the supp. Mat. A uniform model labelling would be appreciated.

We have added a line at the top of the table explaining what is calculated concentrations and what is correlations. Model labeling corrected.

Table 2: ibid about CHASER_rel. The labels of regions should be consistent with Fig 1.

Labels for regions now as in Figure 1.

Table 3: swap the first and second sentences. Use boldface rather than larger signs

for important deviations.

First and second sentence swapped. Large + and - replaced by bold face.

Figure 2. GFDM_AM3 missing for CO_Tracer

The calculation of RERER require BASE, GLOALL and EURALL. EURALL calculations are not available from GFDL_AM3.

Figure 3: panels g) and i) appear identical

g and i now corrected.

Figure 4 and 5: Truncation fixed

Figure 6: add that the results to 20% perturbation are plotted.

We have added that the the GLOALL scenario is calculated with with 20% reductions in anthropogenic emissions.

Supp. Table 1: Table 1: all models are in bold, not the first 7, since 7 models are displayed. The information about spin up should be given in the experiment description, not in Table 1. Footnote #2 is not references in the text.

We have removed bold face for the last model. The information about spinnup now in the experiment description. Footnote 2 deleted

Supp. figure 1: What happened in Heimaey in May in observations?

The peak is actually the IFS model. We have changed the colour scale for the models so that it is easier to tell models and measurements apart. In the IFS model it could be an artefact of the GFAS fire emissions (April/May 2010 is the time of EEyjafjallajökull eruption but it is not clear how this can effect CO).

Supp. figure 4: More details are needed in the legend about the source of data and the indicator displayed.

More information about data source added. Editing errors corrected.

---

## Author Comment (AC2) · 8 Jun 2018

We thank the reviewer for the effort to see into this multi-author paper. We apologise for oversights, partly due to the complex nature of the multi-model evaluation.

Reply to general comments:

Most of the general comments are addressed in the detailed comments below, eg error in figure 3, fig 3i, AM reference etc.

At the time of the submission we believed that another paper would address the validation of ozone. As it turned out this was not the case, and as a result the ozone validation has been extended in this paper. We have reorganised the paper. The content of in particular section 5 has been expanded, and a motivation for this section is

[Figure]

included in the introduction.

The issue of ozone titration is discussed in more detail. The reviewer is right that as in particular European NOx emissions have decreased from 2001 to 2010, and as a result European controls have been offset by by removal of local suppression. These considerations are now discussed in section 4.4.

Detailed comments: ——————

Line 16: capitalization error.

Comma replaced by .

Line 25: Missing parenthesis

Added right )

Line 29: Verb agreement. Replaced is with are.

Line 46-54: The list of published papers should be used to provide context. Here it is simply a list.

We have deleted this list and replaced it with this: A large number of papers from HTAP2 have been published in the ACP (Atmospheric Physics and Chemistry) special issue: "Global and regional assessment of intercontinental transport of air pollution: results from HTAP, AQMEII and MICS"

Line 60: Details like model count would be better in the methods.

Model count deleted here.

Line 68-70: Differ should be differences?

Not applicable. The description of the sections later in the paper is changed.

Line 72-74: Poorly written.

Now corrected to: The HTAP2 model experiment was set up by the Task Force on

Hemispheric Transport of Air Pollution (TF~HTAP). A project work plan, a description of the model experiments etc can be found on the TF~HTAP web page (\url{ http://www.htap.org/}).

Line 82: "etc" seems particularly poor when later you will refer to advection schemes as a causal difference.

Advection added to list, and a reference to the supplementary material. These models have different resolutions, advection schemes, chemical mechanisms etc (see supplementary material and references therein).

Line 87: Space added

Line 92: Space added

Line 95: Replaces in by is

Line 95: How does evaluation of upwind sources affect conclusions about transport to Europe?

We have not included an evaluation of upwind sources here. Several other HTAP2 papers are addressing this.

Line 99: GAW (Global Atmospheric Watch) spelled out.

Line 101: How "high" correlations are expected given the resolutions of the models?

We have included some more text and references here: Correlations shown here are in the same range as correlations with MOPITT satellite measurements as reported by Naik et al.2013. However, as shown in Table 3, all models except IFS\_v2 underestimate annual CO levels by 13% or more. Similar underestimations was also shown Strode et al. 2015.

Line 101-102: resolutions of all the models should be provided in the methods rather than the comparison to measurements.

The information about the resolution of the CHASER models is provided here as explanatory information for how the versions of the CHASER model differ. Information on model resolution for all models is given in a table in the supplementary, referenced in section 2.

Line 98-109 How is it that CO deserves a site-by-site comparison and ozone?

Unfortunately it was communicated to us until a few days before the manuscript had to be submitted that this would be included in the Galmarino et al. paper. Therefore is was not included in this ACPD submission. We are now including a site-by-site surface ozone evaluation based on GAW data as already included for CO.

Line 112. The authors should mention that they do have some surface evaluation in this paper. Currently, Table 3 in this manuscript is not referenced until Section 5.

We now say that there is additional surface evaluation in the Galmarini paper, and we refer to Table 3 also in this section and too all supplementary material on ozone evaluation.

Line 114: There is currently no discussion of ozone results except to say they exist somewhere in the supplement. Why is this sufficient?

We have added a more complete paragraph on the ozone comparisons made in chapter 3.

Lines 122 - 123: There must be more discussion of the basic results that will clearly affect transport.

This section has been extended and now reads: The profile comparison allows to identify differences between the models in vertical mixing of ozone useful for further interpretation in inter-hemispheric transport efficiency. Note that the GEOS-Chem model only simulates ozone in the troposphere and its ozone levels above 300 hPa should be disregarded. With a relatively inactive chemistry in the winter months the measured ozone profiles show little vertical variability, with ozone mixing ratios in the troposphere

increasing gradually with height. Model calculated ozone profiles are in general close to the measurements. As the chemical activity increases in Spring and summer months the vertical variability increases, reflecting air masses of significantly different photo-chemical history at different levels. As was shown in \cite{Jonson2010} the models are not capable of reproducing this vertical structure in ozone levels. Most of the models underestimate free tropospheric ozone in the summer months.

Line 138: Here and elsewhere the definition of regions is incorrect. Here you have NW, SW, SE, GR+TU. In the Figure, you have NW SW, E, GR+TU. Other places you have NW, SW, E, SE. Choose one, and be consistent.

The region notations are now consistent throughout the paper.

Lines 139 - 140: Is this source apportionment the same as contribution in sections 4.4 and 4.5?

Yes. We have now included references to the subsections.

Line 142: rate of decay is later explained, but here seems completely arbitrary.

We disagree. We think that the rate of decay is useful information/reminder here.

Line 182: Numbering of Figure 3 and 4 corrected.

Lines 185 - 189: The reasonableness of this should be discussed.

Differences between the individual models are very similar for CO and the CO tracer. Differences in the CO tracer can only be caused by advection as there is no chemistry for this species. The similarity between CO and the CO tracer for two models leads us to believe that the causes for the differences are the same.

This argument is included in the text.

Line 205: This gets discussed in several places and is really part of the methods.

OK, shortened here, but this information is also repeated here as part of the interpre-

tation of the results.

Lines 217 - 2019: Web citation is inappropriate. Further, the lifetime of ozone is expected to vary with respect to season and altitude (Wang et al. 1998; Brasseur, Orlando, and Tyndall 1999). Estimates of lifetime at 500hPa range from 15-160d and from 40-300d at 10km. Your upper bound of 18days is misleading. Table 1.1 of the HTAP 2010 report cites weeks to months in the free troposphere. The IPCC range of values do not acknowledge the complexity of ozone transport.

In acknowledgement of the complexity of ozone chemistry and transport, we now refer to the HTAP 2010 report for the lifetime of ozone. In addition we have replaced the web citation with a reference to the IPCC report.

Line 242: AM3?

The GFDL_AM3 model is added to the list of models not perturbing aircraft emissions.

Lines 246 - 247: Provide some reference or evidence.

We are now referring to a paper by Cameron et al. (2016} for the effects of aircraft emissions on surface ozone calculated by several global models. See updated discussion in the manuscript for details.

Line 247: here = PBL?

This part has been rewritten.

Lines 254 - 284: Is this contribution from a simple mean within seasons? What months were included in each season? Are the numbers in the text ensemble means? What about ensemble mean RBU? MDE? EU? 290-291: Did they "too" calculate smaller "than in this study" or did they "too" calculate "smaller as in this study"?

This section has been rewritten. See also comments from reviewer 1. In Figure caption 5 we now specify which months are included in WI, SP, SU and AU. 0.37 NA to EU and 0.17 EA to EU are from Table 4.2 in the HTAP1 report. The numbers are ensemble

means. This is now noted in the text. We have chosen not to compare the numbers for EU as the definition of the European domain is so different. We now also list list the numbers for the remaining regions. They are also shown in Figure 5.

Line 269: MDE appears to always be small.

We now say that contributions from the Middle East and North Africa are small.

Lines 290 - 291: Did they "too" calculate smaller "than in this study" or did they "too" calculate "smaller as in this study"?

The text is changed to make this clearer: They calculate a much smaller contribution from non European sources than in this study, similar to the contributions calculated in HTAP1.

Lines 272 - 273: Did these other studies use the same model?

We have added that in Jonson et al. (2015) the EMEP model was used. Brandt et al. (2013) used a different model.

Line 277: Right parenthesis added.

Lines 274 - 280: Methods?

This part is now rewritten.

Line 305: HTAP1?

We have added that the Fiore et al. paper was based on the HTAP1 model experiment.

Lines 306 - 335: There needs to be a clearer connection to the previous section. In fact, you could just add two bars to Figure 5a. That would help to connect the of POD and SOMO35 to the seasonality of titration.

We have added more material to this section following the recommendations also from reviewer 1

Line 361 - 371: Terse and uninformative

This section is rewritten bringing more information.

Line 390 - 392: See previous comments about ozone lifetime.

We now refer to discussions on lifetimes in previous section.

Line 400: Probably deleted. Improved text.

Line 405: added for ozone.

Comments regarding Table 1: If mountain sites are used at readers peril, consider making room for ozone evaluation by moving them from the first data result.

We have included a similar table as Table 1 with ozone. We have not included mountain sites in the ozone table as the "peril" is much larger for ozone as the dry deposition is faster and lifetime shorter.

Comments Table 2: Update region definitions to be consistent with figures and text.

Regional definitions updated.

Figure 1: update region names to be consistent. Also, too many extra colors so it is hard to tell what is included. Is the Baltic Sea part of Eastern Europe? Black Sea? Caspian Sea? Mediterranean?

Not changed. Difference in colour is visible both on the screen and on printout. The European seas are part of the OCN region.

Figure 2: Necessary?

We would definitely like to keep the figure. We think it illustrates very well the evolution in RERER going from a simple CO tracer to CO and finally ozone with a multi model ensemble.

Figure 3: lettering needs to be updated in the figure and in the text. What was the common grid and how was it treated when a grid cell at 1000hPa was below the surface?

Lettering is updated and is now the same as in the text. All model data has been interpolated to a common vertical grid. For gridcells below 1000hPa values at the lowest model level was used.

Figure 3: 3i is AM3 CO not ozone. Column 3: consider a scale that does not saturate in so much of the figure.

Panel 3i corrected.

Figure 4: North and south boundaries are unnecessarily different from figure 3. Further, this highlights that no meaningful discussion of the boundaries was made. In fact, 50E includes a lot of Russia and a lot of ocean. Column 3: consider a scale that does not saturate in so much of the figure.

North and south boundaries changed corresponding to Figure 3. This resulted virtually no visual changes in the figures. We have added a discussion on the boundaries: This area roughly corresponds to the European regions as shown in Figure 1, but also some additional land and sea areas. The main focus of the figure is in the free troposphere where horizontal gradients in concentrations are small. Liu et al. 2009} calculated the correlations between nearby pairs of sonde stations. They found low correlations near the surface indicating that local and regional effects are important here. From the surface correlations rose sharply to a local maximum in the lower troposphere. We therefore conclude that the selected area is a good representation of the atmosphere above Europe.

Figure 5: There is no discussion about the CHASER model being the only one without apparent titration, and this should be discussed somewhere. Region definitions should be consistent with the text or the text should be consistent with the figure. The units are cutoff on the first row.

We have commented the low level of ozone titration for the CHASER model in section

4.3: For all models, except the CHASER\_re1 model, ozone titration dominates the overall European contributions when summed up over the three winter months. However, for all the models, including also the CHASER\_re1 model, the net European contributions includes regions of net ozone production and net ozone destruction in winter.

Regional definitions now consistent with text.

Figure 6: Region definition nomenclature. I recommend showing as 3 stacked-bars (or adding to Figure 5). If I am interpreting this right, the RAIR is 84% compared to 43% from HTAP1. I suspect that all models provided annual and I think reporting RAIR would be useful (maybe in Figure 2).

Region definition nomenclature fixed.

Figure 6 is complemented by a table with results for summer ozone from the models following the recommendations from reviewer 1. For ozone this table lists the annual (and summer) percentage contributions to Europe from several regions, including Europe to it selves. We have also calculated average RAIR for the models in Figure 5. The HTAP2 RAIR of 82% compared to 43% in HTAP1 is discussed in section 4.4, and these numbers are also repeated in the conclusions. RAIR for the individual models proved difficult with European contributions to it selves was close to zero and even negative for some models.

---

## Author Response (AR2)

Dear Editor

Below follows a point-by-point reply to the comments raised in the review (answers to the comments in italics). We have also received several comments from one of the co-authors (Frank Dentener). We think that his comments have added to the quality of the paper. As a result there are more changes in the manuscript than requested in the review. Changes related to the revision and the additional changes (mainly suggested by Frank Dentener) are included in the marked-up manuscript below.

**Abstract:** More quantitative results from the conclusion should be included in the abstract. In particular the much lower contribution of European emission than found in HTAP1 and the subsequent increase of RAIR.
*More quantitative results have been included in the abstract on the HTAP1 and HTAP2 comparison, and the reasons why the results in HTAP2 differ from HTAP1.*

**Section 3.1:** Confirm the suspicious seasonal cycle at Rucava, Latvia (Supp Mat. Fig 4).
*Unfortunately Rucava was mistakenly replaced by a different site. The correct Ruccava figure is now included.*

On the same figure: are the observations for Pallas missing most of the year ?
*Pallas is removed as there are no measurements for this site.*

What is the temporal frequency on the scatter plots (Fig 7 of the Supp Mat), the legend says those are annual data, but it is surprising to see so many annual means, i.e. 1302 European stations for the EMEP network.
*This was an error on some of the plots, that has been corrected. Data are monthly (ca 1300 in that year) and are from 113 stations in Europe.*

**Conclusion:** the lower impact of European emissions compared to HTAP1 is the most important result of the study (and perhaps of the whole HTAP2 project), it would deserve a few more words. Besides the actual sensitivity provided in L561, the standard deviation provided in the author response should also be included to support whether that signal is still significant. I am also missing here more conclusions on the importance of shipping, which is now isolated from the European signal (which was not the case in HTAP1) and to what extent this may also contribute to a lower role of European emissions compare to the previous assessment.
*In the conclusions we have now also listed the effects on ozone in ppb induced by the 20% changes in North American and East Asian from HTAP1 and HTAP2. Uncertainty ranges given in section 4.4 are included in the conclusions. The effects of including OCN and RBU as separate HTAP2 regions as oposed to being parts af a bigger HTAP1 European region are discussed in the conclusions.*

**Technical comments**

P2L30: CTMs are chemistry transport models

OK. Fixed

P4L105: the word now in model data are now included in the database is not appropriate with regards to the (long term) temporality expected from a peer reviewed paper.
OK. The word now deleted.

Supp. Matt. Figure 4 and 5 : the labels of the panels (a,b,c) needs to be fixed.
OK. Fixed

P17L561: itself.
OK, fixed

P19L643: correct the acknowledgement.
OK. Fixed

[revised manuscript text omitted]

**Figure 5.** Contributions to European ozone levels (in ppb) from different world regions. (WI is December, January, February. SP is March, April, May. SU is June, July, August. AU is September, October, November). Note that the separate contribution from North Africa (NAFALL) and ocean shipping (OCNALL) is only included in the EMEP_rv48 model calculations. The Middle East (MDEALL) and Russia, Belarus and Ukraine (RBUALL) is not included in the IFS_v2 model. For all models contributions from missing regions are included as ROW (rest of the world). Note that the areas included in ROW is model dependent. For the four top row models the effects of a 20% increase in $CH_4$ is shown as a separate bar.

[Figure]

**Figure 6.** Contributions to ozone metrics annual mean ozone, SOMO35 and $POD_1$ forest in percent (where BASE - GLOALL represents 100%) as calculated by the EMEP_rv48 model. The metrics have been scaled so that the difference between the the BASE - GLOALL (20% anthropogenic emission reductions) calculations is 100% (the sum of EUR, NAM, EAS, RBU, OCN and ROW adds up to 100%).